# A Structured Narrative Prompt for Prompting Narratives from Large Language Models: Sentiment Assessment of ChatGPT-Generated Narratives and Real Tweets

Christopher J. Lynch [1,*] , Erik J. Jensen [2] , Virginia Zamponi [1] , Kevin O'Brien [1] , Erika Frydenlund [1] and Ross Gore [1]

1   Virginia, Modeling, Analysis, and Simulation Center, Old Dominion University, 1030 University Blvd., Suffolk, VA 23435, USA; vzamponi@odu.edu (V.Z.); kobrien@odu.edu (K.O.); efrydenl@odu.edu (E.F.); rgore@odu.edu (R.G.)
2   Computational Modeling and Simulation Engineering Department, Old Dominion University, Norfolk, VA 23508, USA; ejens005@odu.edu
*   Correspondence: cjlynch@odu.edu; Tel.: +1-757-686-6248

**Abstract:** Large language models (LLMs) excel in providing natural language responses that sound authoritative, reflect knowledge of the context area, and can present from a range of varied perspectives. Agent-based models and simulations consist of simulated agents that interact within a simulated environment to explore societal, social, and ethical, among other, problems. Simulated agents generate large volumes of data and discerning useful and relevant content is an onerous task. LLMs can help in communicating agents' perspectives on key life events by providing natural language narratives. However, these narratives should be factual, transparent, and reproducible. Therefore, we present a structured narrative prompt for sending queries to LLMs, we experiment with the narrative generation process using OpenAI's ChatGPT, and we assess statistically significant differences across 11 Positive and Negative Affect Schedule (PANAS) sentiment levels between the generated narratives and real tweets using chi-squared tests and Fisher's exact tests. The narrative prompt structure effectively yields narratives with the desired components from ChatGPT. In four out of forty-four categories, ChatGPT generated narratives which have sentiment scores that were not discernibly different, in terms of statistical significance (alpha level $\alpha = 0.05$), from the sentiment expressed in real tweets. Three outcomes are provided: (1) a list of benefits and challenges for LLMs in narrative generation; (2) a structured prompt for requesting narratives of an LLM chatbot based on simulated agents' information; (3) an assessment of statistical significance in the sentiment prevalence of the generated narratives compared to real tweets. This indicates significant promise in the utilization of LLMs for helping to connect a simulated agent's experiences with real people.

**Keywords:** narrative generation; simulation; large language models; natural language generation; ChatGPT; structured prompt; prompt engineering; prompt design

## 1. Introduction

Policy makers, decision makers, and researchers operating across ingroup–outgroup settings are challenged with understanding and empathizing with choices and events within systems that do not necessarily reflect their own life experiences. Policy makers need ways to peer into these systems without their social status being reflected back at them. Simulation provides groups perceived as outsiders with in silico viewing portals into systems of interest, such as marginalized communities, low-income areas, displaced communities, etc. The process of using simulation to explore real systems involves pipelines such as collecting real data from real individuals, training targeted models to reflect the system, generating simulation data, assessing outcomes, and then cyclically re-engaging community members for new information and progressing back through the pipeline with

new research questions and agendas. This may yield useful information but lead to *over-researched* fatigue and participation resistance by community members [1,2]. As a possible avenue to mitigate these concerns, researchers and policy makers can lean on simulations to reflect systems of interest populated with simulated agents. The information created in these simulations can allow for a better understanding of the context of the system and its occupants, and help guide towards more informed participatory simulation designs [3]. Large language models (LLMs) can play a promising role in the ability to create useful narrative messaging that reflects both the emotional state of the individual that a narrative is based on and the context of the environment. To this end, the use of narrative messaging can be powerful in informing opinions and decisions by appealing to emotional and social contexts through integrating characters, action, and plot [4,5]. These narrative elements can be further informed based on the content of the simulated setting, agents, and contextual information, such as the social norms associated with the system.

Appropriately understanding the social, societal, emotional, and ethical components of simulation models developed to represent societies, societal issues, or behaviors of individuals is a challenging task. This is particularly true in agent-based models (ABMs) where individuals and individual behaviors, interactions, and decisions are modeled and allow for the aggregate system behavior to be explored through the interactions of the individual agents over time [6–9]. Understanding outcomes as well as the paths that led to these outcomes is muddied by the need to track changes over time (i.e., interactions, histories, and changing individual perceptions), understand the social norms of the system, and filter through large volumes of data [10,11]. Many engineering-based solutions exist for exploring ABM outcomes in ways that facilitate understanding and provide data support findings, such as statistical debugging [12,13], visual inspection [14,15], or logic tracing [16,17]. However, these approaches generally (1) convey localized information about the system that is restricted in scope, (2) require domain knowledge of analytics in addition to the modeled system in order to facilitate proper understanding, (3) sometimes provide statistical significance in support of outcomes, (4) do not inherently or intuitively connect the content of the outcomes with the practical significance for the policy makers, decision makers, and/or researchers utilizing the results, and (5) are very time-consuming to explore [18–20].

Currently, LLM-based chatbots such as ChatGPT are of great interest to the scientific [21], medical [22–25], and engineering [26,27] communities for research and education, and in clinical settings, for easing provider workflows and improving patient outcomes [28–34]. Utilization of LLMs for these purposes already demonstrates many benefits and LLMs promise to deliver additional benefits as the technology improves, and as more tools are developed for specific domains. Critical concerns with using LLMs for any task include accountability, safety, responsibility, and enforcing honest use [24,25,35]. Chatbots have been shown to provide erroneous outcomes, or hallucinations, that they convey in an authoritative manner, which may mislead users [28,36]. Many articles highlight the possibilities and intricacies associated with the integration of LLMs and reinforce the importance of understanding the training data for the LLM, balancing LLM suggestions with practical domain knowledge, assessing accuracy and bias, and maintaining security and accountability [25,29–31,37–39].

We posit that individualized narratives can convey relevant information to model users that captures the emotional, social, and societal state of the simulated agent in a manner that can better resonate with the user. LLMs can capitalize on the large volumes of data generated from the perspectives of the individual agents, as well as that of the aggregated system, to communicate unique, personalized messages about key life events, interactions, or changes to the agents' well-being, cognitive or emotional states, or level of happiness. The capability of LLMs to generate stochastic and wide-ranging responses can increase the diversity of thought represented in ABM agent narratives.

Typically, user input into an LLM chatbot takes a conversational form. However, this kind of verbose, natural-language format is not convenient as an intermediary between the

ABM, which generates agent and event data, and the LLM. Therefore, it is optimal to input the ABM data into a structured prompt with formulaic fields for the LLM to interpret. No structured prompt has yet been developed for consistently making requests of publicly available LLMs to generate narratives based on simulated agents. Therefore, to facilitate reproducibility, transparency, and reuse of a process for generating narratives based on simulated agents, we develop a structured narrative prompt that combines the desired information comprising a narrative (i.e., the subject of the narrative, the type of life event being described, the age of the agent being represented, etc.) with the target audience for the narrative (i.e., a policy maker for a specific community or locality, a lay person on social media, an academic researcher, etc.). While the LLM structured prompt design used in this work is specific to ABM narrative generation, we suspect that our work, specifically the development of the structured prompt and our findings on ChatGPT's capability to interpret formulaic data, is of interest to other fields in which it is desirable to submit formulaic LLM prompts, e.g., in healthcare [39].

Initial testing with the LLM ChatGPT's manual web interface demonstrated that the process of creating naturally digestible narratives using simulated agents' information was promising. However, slight variations in requests for the narrative resulted in drastic changes to the generated narrative. Many narratives (1) contained inaccuracies with respect to the provided information, (2) provided additional information within the story that was not provided as an input, or (3) expanded the narrative in ways that were not requested and were not possible for the LLM to know. These initial findings were not unexpected and are in line with other assessments of LLMs for natural language processing (NLP) tasks [40,41]. As a result, we developed a structured process for prompting LLMs and tested our structured narrative prompt using OpenAI's GPT-3.5 LLM via the API [42].

We conduct hypothesis testing of positive and negative sentiment comparisons between simulated agents' narratives created using the GPT-3.5 LLM and sentiment contained in real tweets collected from Twitter [43], now known as X. Sentiment comparisons are conducted using the Positive and Negative Affect Schedule (PANAS) traits for describing feelings and emotion [44,45]. We are interested in the areas where the sentiment contained in the ChatGPT-generated narratives in statistically significantly indistinguishable from the sentiment contained in real tweets. Narratives are created using simulated agents' life events themed on births, deaths, hirings, and firings, and are assessed for sentiment prevalence across 11 PANAS categories for a total of 44 outcome categories (four event types and eleven sentiment traits). Overall, four categories were found to have differences that were not statistically significantly different between the ChatGPT-generated narratives and the real tweets. Additional work is needed to assess the validity of the generated narratives, which is the focus of follow-up research.

This article provides three contributions: (1) a structured narrative prompt digestible by LLMs for generating narratives from simulated agents' life events and information (described in Section 3.1); (2) a list of LLM benefits and challenges in research (described in Section 2); (3) an assessment of statistically significant differences in sentiment scores between ChatGPT-generated narratives and real tweets (described in Section 3).

## 2. Materials and Methods

We explore the use of LLMs to generate narratives based on any life events deemed interesting, impactful, or key for simulated agents. Prior advances in the field of producing realistic narratives based on simulated agents' life events have focused on generating and posting tweets based on real-time events within a simulation based on the current state of mind of an agent at the given point in time [46]. This prior work developed a framework for creating narratives based on individual agents' life events and posting tweets during the simulation runs [46]. Their goal was to provide processes through which empathy could be generated for simulated entities through a variety of communication mediums while maintaining a connection to the agents' data, decisions, connections to other agents, and their histories. That work relied on the creation of numerous Java classes to

form narrative frameworks that could be populated with an agent's pertinent information. The outcome yielded interesting individual narratives generated in real time, but with a rather scripted and recurring feel when observed as a batch of narratives.

We utilize ChatGPT's API to generate natural-language-sounding narratives and we utilize real tweets to compare sentiment levels across PANAS traits. The narratives generated using ChatGPT incorporate information generated and collected throughout a run of an ABM. The components that are captured from the ABM include an event type (birth, death, hired, or fired) and information deemed necessary to generate a relatable narrative for the given event. This includes information such as the agent's name, location, ID, relationship to the subject of the event (i.e., "self" if referring to a hiring or firing event or the name of the corresponding agent if describing a birth or death of another agent). Sets of narratives are generated around each of these four event types. The tweet set is not categorized along any specific event type. The set of tweets is used holistically to compare sentiment levels against the tweets within each life event type from ChatGPT-generated narratives.

Figure 1 depicts the process of generating life event information that can yield a narrative, deriving narratives from the life event information, and assessing sentiment levels within each narrative. The life event information is generated using an agent-based model (ABM) that generates "life" events, including births, deaths, job hirings, and job firings. This work takes the event and corresponding narrator and subject characteristics, generated by the ABM, and uses a large language model (LLM) to generate one or more LLM narratives. This event information is organized in a formally defined prompt structure for input into the LLM. The yellow boxes in Figure 1 show the flow for ABM event and LLM narrative generation. The generated LLM narratives are intended to be similar in function and style to tweets that have been posted by real people.

The blue boxes shown below the gray dashed line convey the flow for real tweet generation from real-life events. For each LLM or human-generated narrative (tweet), PANAS sentiment analysis generates a set of binary values indicating the presence or absence of key category-specific dictionary words in the narrative, as shown in the red box in Figure 1. Statistical analysis on these binary sets can differentiate LLM narratives and human-generated narratives (orange box), based solely on the binary presence of PANAS category keywords. Sentiment and statistical analyses are discussed further in Section 2.1 and diagrammed in Figure 2.

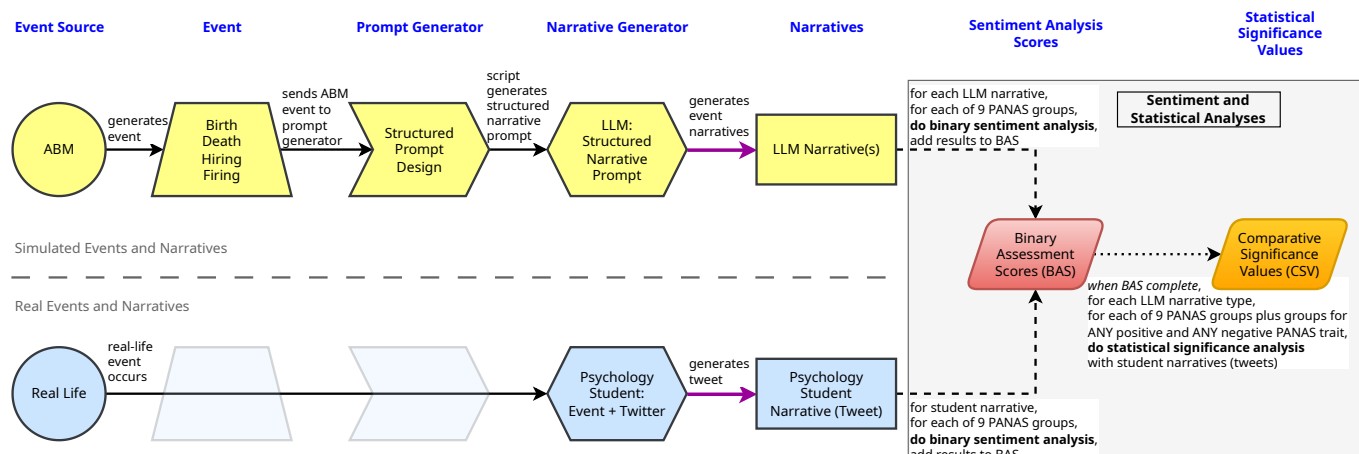

**Figure 1.** Process flow for taking simulated event data, generating narratives, and performing sentiment analysis and statistical comparisons between narratives and real tweets.

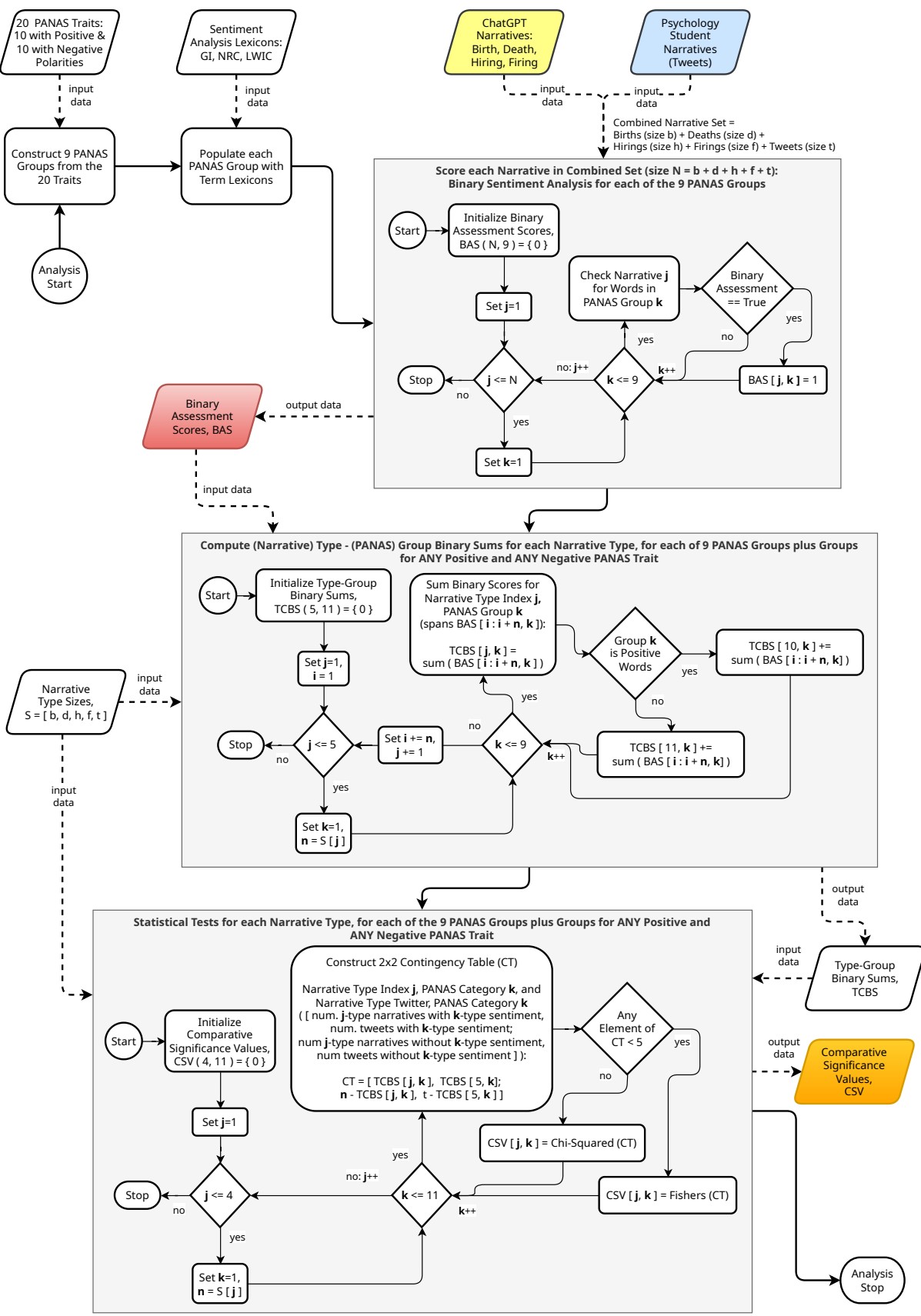

**Figure 2.** High-resolution process flow for performing sentiment analysis of Boolean PANAS trait groupings and statistical analysis of significant differences between ChatGPT-generated narratives and real tweets.

Numerous applications of using LLM-based chatbots exist in the medical, scientific, and engineering communities. These reports and studies yield valuable insights into how LLM-chatbots, such as ChatGPT, have been beneficial and challenging across a variety of uses. Table 1 summarizes these demonstrated and proposed benefits and challenges. The identified benefits support that LLM chatbots perform well when given multimodal [30] and validated information [36] and that consistent and well-structured outcomes can be created [27]. The identified challenges share themes of concerns over validity, uncertainty, bias, and accountability when creating or utilizing LLM outputs. To this end, our structured narrative prompt is developed with these challenges in mind, and is intended to add value to the existing body of knowledge by providing transparency, consistency, and traceability to the process of transforming agent data into natural sounding narratives.

**Table 1.** Benefits and challenges of using LLMs as tools for research, medical, clinical, training, and engineering tasks.

| LLM Benefits | LLM Challenges |
|---|---|
| <ul><li>Efficiency and efficacy in doing **medical research**, including summarizing literature and identifying research gaps [29–31]</li><li>Assistance in **clinical settings** with note-writing, patient inquires, and data management [28,29]</li><li>Facilitation of **scientific exploration**, and assistance with **research activities**, including experimental design, peer review, and grant applications [29]</li><li>**Clinical decision** support, e.g., for developing testing and treatment strategies [28–31,37,39]</li><li>Improving **diagnostic accuracy** and predicting **disease progression** [37,39]</li><li>ChatGPT can produce responses to medical inquiries that are superior in **quality and empathy**, compared with similar social-network doctor responses [30]</li><li>**Domain-specific LLMs** fine-tuned with clinical data such as electronic health records, and **novel architectures** that can integrate longitudinal and multimodal data [30]</li><li>**Collaboration** with medical experts, ethicists, data scientists, policymakers during model development [37]</li><li>Model **validation** with medical experts in clinical settings [37]</li><li>Potential for **fairness and equity** in healthcare for diverse populations [37,47]</li><li>Interactive and personalized **education and training** for medical students and practitioners [29–31], and engineering students [48,49]</li><li>Potential for revolutionary new **biomedical engineering** LLM tools [33,34]</li><li>Support for many languages and **global** access to medical knowledge [37]</li><li>Internet-connected models with access to new research can deliver **up-to-date** information [30]</li><li>In US and EU, LLMs that perform medical decision-making tasks are classified as medical devices and are **regulated** [36]</li><li>Training medical LLMs with only controlled and **validated text** improves capabilities [36]</li><li>Boosting **timeliness and volume** in idea generation [27]</li><li>Developing domain-specific requirements tables in SysML that are **well structured and consistent** [32]</li></ul> | <ul><li>**Reasoning errors** or chatbots' inabilities to critically evaluate and link cause-and-effect relationships [21]</li><li>Tendency for LLMs to "hallucinate" and provide **convincing but incorrect** responses [36,50,51] including invented references and inaccurate dates [21]</li><li>Lack of **transparency** of data sources and data providence for LLMs like ChatGPT [24,50]</li><li>In US and EU, LLMs that perform medical decision-making tasks are classified as medical devices and are **regulated** [36]</li><li>Medical LLMs can be and have been trained with **inappropriate** and/or **outdated** material [36]</li><li>Difficult to **validate** LLMs designed for critical tasks: safe use should require understanding of relationship between LLM **inputs and outputs**, **bounding** of LLM outputs to only correct information, a guarantee that successful testing proves accuracy of **future results** [36]</li><li>Prevention of the **racial and gender biases** that have been well-documented in non-LLM AI tools used for medical, policing, and surveillance tasks [52–56]</li><li>**Gender bias** in LLMs like ChatGPT that reflects cultural gender stereotypes [57]</li><li>Biases in scientific writing in the Humanities, including language bias favoring English sources, **neo-colonial bias** favoring Western authors, and **citation bias** tending towards older and more highly cited publications [21]</li><li>Ethical and practical concerns of using LLMs trained on **biased** data for critical tasks [24,29–31,37,58]</li><li>Current LLMs are unable to estimate **uncertainty** of responses [36], but uncertainty quantification is an active area of research [59,60]</li><li>Problematic to guarantee the domain-specific **accuracy** of LLM training data and responses [29–31,58]</li><li>Difficult to **interpret** how an LLM is processing data and making decisions [30,61]</li><li>Risk of clinician **over-reliance** on AI tools [37]</li><li>Maintaining **accountability** of clinicians who are making decisions using LLM tools [25,30,31]</li><li>Ensuring **security and privacy** of patient data used for training and in clinician prompts [29–31,37,39,50]</li><li>How to manage **integration** with clinical experts for development and validation [30,31,38]</li><li>Designing practical and effective **feature-based** prompt structures for clinical settings, as opposed to using narrative prompts [39,41]</li><li>Nuance of **medical language** and **context diversity** may be difficult for LLMs to capture [37]</li><li>Impaired model validation due to **contamination** of training data with testing data [38]</li><li>**Public distrust** of and dislike of AI technology in general due to concerns about plagiarism, misuse, environmental impacts, training-material misappropriation, existential threats, etc. [25,29,50]</li><li>Need for **monitoring and assessment procedures** to ensure that pilot studies and experimental projects account for ethical and social practices for the human and not just for commercial and prestige interests at the introduction point of an AI system as well as over the life span of the system [61]</li></ul> |

Prior works have relied on templated Java classes to create unique yet consistently structured narratives that contained structured responses populated with data specific to the agent, such as names, data values for age, income, etc., location information, or job

information [46,62]. This approach required disproportionately large volumes of Java code to be written for the small number of scripted responses that could be generated. We flip the perspective to create a structured narrative prompt that can be used to send information to an LLM and get back a wide variety of responses without having to modify code structures based on any specific event type, simulated agent type, or target audience. All of this information can be wrapped inside the structured narrative prompt for consistency and transparency.

*2.1. Sentiment Scoring and Statistical Analysis*

Narratives communicating important information for an agent, such as an associated birth or death, should follow a realistic flavor for the given environment while maintaining a flavor that is unique both to the agent and the agent's current state of mind. Therefore, the sentiment conveyed in a narrative should vary depending on the type of life event. A birth announcement may be themed in a very joyous manner while a narrative pertaining to a death may be much more melancholy and reminiscent. We aim to assess whether statistically significant differences exist in feelings and emotions between narratives generated by LLM platforms and real tweets with respect to PANAS groups. We generate narratives framed around Birth, Death, Hired, and Fired events for comparative evaluation between the sentiment of real tweets and the sentiment of the generated narratives.

An archive of tweets generated by psychology students [43] is used as a corollary to the LLM narratives to establish a baseline sentiment value for comparing against the generated narratives. Note that the ChatGPT-generated narratives are based on four specified event types while the real tweets covered a wide range of unrelated situations over a one-year period. All of the tweets are compared to each set of generated narratives for each event type, thereby providing a static sentiment level for comparison across each event type. Our evaluation relies on the large, unbiased vocabulary present within the tweet set for enabling the generalized assessment of sentiment scores across the 20 PANAS categories and facilitating assessments for statistical significance.

The statistical tests selected include the chi-squared test and Fisher's exact test. For both tests, it is assumed that the data can be divided into a two-by-two contingency table with two groups (i.e., LLM-generated narrative dataset versus real-tweets dataset) and two categories (e.g., narratives demonstrating any amount of positive sentiment versus narratives demonstrating zero positive sentiment). We then apply Fisher's exact test or the chi-squared test to look at whether significant differences in sentiment values exist between the PANAS groups for the given comparison categories. Fisher's exact test [63] is appropriate when very small sample sizes exist within any of the cells of the contingency table [64]. To determine which test to apply in each case, we assess whether any cell of the two-by-two contingency table has fewer than five samples. If any cell has fewer than five samples, then we apply Fisher's exact test; otherwise, we apply the chi-squared test. For the chi-squared test, we provide the $\chi^2$ test statistic, the degrees of freedom (df), and the $p$-value. For Fisher's exact test, we only provide the $p$-value. Fisher's exact test has been used in recent works to identify urban functions [65]; to isolate differences between pregnant women with and without pregnancy-induced hypertension [66]; to find differences in crocodile lizard populations regarding parasitic infection [67]; to identify associations between single-nucleotide polymorphisms with autism spectrum disorder patients [68]; and to perform analyses of the development of lower-back pain in school-age children [69]. Figure 2 provides a high-resolution depiction of our sentiment scoring and statistical analysis algorithms. The color scheme in Figure 2 mirrors the color scheme used in Figure 1.

To score the sentiment of each message and allow for comparisons between narrative sources, we utilize the PANAS lexicon. The PANAS lexicon reflects words associated with the well-validated Positive and Negative Affect Schedule (PANAS) [44,45]. PANAS consists of 20 traits, with 10 traits measuring positive affect (e.g., excited, inspired) polarity and 10 traits measuring negative affect (e.g., upset, afraid) polarity. These 20 traits are categorized

into nine groups. For each group, a lexicon is formed from categories of words from three canonical sentiment analysis lexicons: General Inquirer (GI), National Research Council of Canada (NRC) Emotion Lexicon, and Linguistic Inquiry and Word Count (LWIC) Text Analysis Tool [70–72]. If the 20 PANAS traits were defined by individual lexicons instead of these nine aggregate groups, then the number of words in each lexicon would be too small to produce reliable sentiment analysis data [73]. The traits included in PANAS, their aggregated groupings, and the specific words in GI, NRC, and LWIC which form each lexicon subgroup are shown in Appendix A.1 [74].

For a given PANAS group, a message (i.e., tweet or narrative) is assigned a value of 1 if it contains any word in the corresponding PANAS lexicon. If it does not contain a word in the specified PANAS lexicon, then it is given a value of 0 for that group. A message with a 1 indicates that the message does contain some sentiment related to the specified PANAS trait. A message with a 0 indicates that the message does not contain any sentiment related to the specified PANAS trait. This is an automated and repeatable approach to sentiment scoring; as such, there is no human or evaluator interpreting the semantic meaning of a message and/or assigning it a sentiment score.

Average scores, standard deviations, and sample sizes are collected for each tweet and narrative with respect to each PANAS group. Sums of 1- and 0-scoring tweets and narratives are tallied for each PANAS group and utilized for statistical significance testing. The null hypothesis is provided in Hypothesis 0 and the alternative hypothesis is provided in Hypothesis 1. The form of the hypotheses remains the same for both the chi-squared and Fisher's exact test. We utilize an alpha level $\alpha = 0.05$ as our cutoff point to determine statistical significance for each test. Calculated *p*-values less than this alpha level provide evidence in support of rejecting the null hypothesis while *p*-values higher than this level fail to reject the null hypothesis in favor of the alternative hypothesis.

**$H_0$.** *There is* no *association between the source of the narrative and calculated sentiment between Group 1 (e.g., ChatGPT-generated narratives) and Group 2 (e.g., real tweets) for a given PANAS category.*

**$H_1$.** *There is* an *association between the source of the narrative and the calculated sentiment between Group 1 (e.g., ChatGPT-generated narratives) and Group 2 (e.g., real tweets) for a given PANAS category.*

*2.2. LLM Selection*

Several LLM-based chatbots exist and are publicly accessible. Google's Bard is a free-to-use chatbot running on the PaLM 2 LLM, available through the web browser and the API. While the original PaLM LLM, which became publicly available in March 2023, was trained on 540 billion parameters [75], PaLM 2 is purportedly trained on a more compact 340 billion parameters, making it more efficient, more performant, and more cost-effective [76]. PaLM 2's training data include "hundreds of human and programming languages, mathematical equations, scientific papers, and web pages" [76]. Bard was previously powered by Google's LaMDA LLM, a conversational AI model capable of fluid, multi-turn dialogue that was fine-tuned using transformer-based neural language models containing up to 137 billion parameters [77].

Meta's LLaMA was created to advance researchers' work in the subfields of AI [78]. LLaMA's initial release consisted of a 65-billion-parameter model which has been expanded to 70 billion parameters with the release of LLama 2 [79]. Currently, researchers and other users must apply for access to LLama 2, which can be run locally only.

OpenAI's Generative Pre-trained Transformer (GPT) models are *pre-trained* with large quantities of source material and use a *transformer* architecture to efficiently *generate* output that is highly dependent on the input [80]. OpenAI offers several different models and two means of access. The ChatGPT web interface with the GPT-3.5 model is free to use, while GPT-4 is available to paying subscribers at a fixed monthly rate. The use of the API is also restricted to paying users but access is prepaid, and funds are debited as a function of prompt and response tokens: OpenAI's text completion models use stochastic sampling of

a set of tokens, which can be words, characters, punctuation, etc., to select the next token in the completion [42]. Similarly, input prompts are deconstructed into a set of tokens, a method of natural language processing (NLP) known as tokenization [81].

The base GPT-3.5 model can utilize a maximum of 4097 tokens, and the base GPT-4 model roughly doubles that token number. Both GPT models are available in large-context form, with about 16k and 32k token maximums [42]. OpenAI trained GPT-3 with 175 billion neural network parameters [82] but parameter information is not available for the newer models. During training, parameters are adjusted to minimize the loss function value which computes the error in predicting the next token in the completion given the context of the preceding tokens [83]. With respect to these identified features and comparisons between LLM platforms, and our prior familiarity with the web portal and API usage, we opted to use OpenAI's ChatGPT-3.5 model to support this research study. Future work may expand to ChatGPT-4, as well as Google's and Meta's LLMs.

### 2.3. ChatGPT API Usage

The OpenAI API documentation offers example code in Python, JavaScript (Node.js), and cURL [84], but the API can be used by any programming language that can make HTTP requests. OpenAI provides an official Python API library which is used for this work. An API key is required to submit prompts. The Python interface offers two functions for submitting prompts:

1. `openai.Completion.create()` is used for single-turn conversations and supports completion models such as `gpt-3.5-turbo-instruct` and `text-davinci-003`.
2. `openai.ChatCompletion.create()` is used for single- or multi-turn conversations and supports chat completion models such as `gpt-4` and `gpt-3.5-turbo` [84].

For the LLM narrative datasets for this work, the `gpt-3.5-turbo` model is used. With the ChatGPT API, it is possible to have multi-turn conversations by submitting prior user prompts and ChatGPT responses with new prompts, up to the limits of the maximum context of the model. Using the official Python interface, prior prompts and responses can be appended to the `messages` parameter in the `openai.ChatCompletion.create()` function call. Any message in the `messages` set is defined by one of three role types:

1. *User*, for application- or API-user-submitted prompts;
2. *System*, for constraints or special instructions that inform an entire conversation, which may be used by software developers to affect the experience of the application user;
3. *Assistant*, for responses to user queries, i.e., ChatGPT responses [85].

Additional function parameters include `temperature`, which affects the stochasticity of the completion algorithm and the range of possible responses, and n, which defines the number of responses generated from a prompt [84]. Multiple responses, for $n > 1$, should be unique if the temperature is greater than zero.

### 2.4. Datasets

The datasets used for our analyses are categorized as: (1) narratives generated from simulation data using ChatGPT; (2) real tweets obtained from Twitter as part of an approved Institutional Review Board (IRB)-exempt study; (3) source codes, the simulated life event data utilized for the narrative generation by ChatGPT, and the PANAS lexicon. All of these components are freely accessible from an online repository [86].

1. ABM simulation output data on Agents' Life Events in CSV format.

    (a) ABM simulation of life event data.

2. Narratives generated using ChatGPT based on simulated agents' life event information. Ten narratives generated per simulation life event.

    (a) Structured ChatGPT API prompts;
    (b) Sets of ChatGPT response narratives.

3. Real tweets obtained from Twitter [43].

(a)     Tweet set with PII removed (dropped IDs and screen names);
(b)     IRB documentation.

4.     Also included:

(a)     Source codes (R): sentiment analysis and statistical significance scripts;
(b)     Source codes (Python): ChatGPT prompt generation, prompt submission, and analysis preparation scripts;
(c)     PANAS sentiment keyword lexicon.

Table 2 breaks down the datasets into the total number of ChatGPT-generated narratives by life event type and total number of tweets. The complete set of simulated agents' life event information generated by the ABM is *filtered* to remove potentially problematic messages, e.g., those with inappropriate narrator ages and poorly defined relationships between the narrator and the subject. From the remaining filtered messages, the *sampled* message sets are generated by random sampling. The ABM generates one narrative from each message, which is included in the study. For each ABM agent's simulated life event, we use the life event type, narrator characteristics, and subject characteristics to prompt ChatGPT to generate 10 LLM narratives.

**Table 2.** Number of ChatGPT-generated narratives and real tweets categorized by life event type.

| Life Event Type | Total ABM Simulated Life Events | Total Filtered Simulated Life Events | Total Sampled Simulated Life Events | Num. ChatGPT Narratives | Num. Tweets |
|---|---|---|---|---|---|
| Birth | 4728 | 4155 | 600 | 6000 | 6148 * |
| Death | 17,340 | 618 | 600 | 6000 | 6148 * |
| Hiring | 26,317 | 3924 | 600 | 6000 | 6148 * |
| Firing | 25,026 | 2860 | 600 | 6000 | 6148 * |
| Real-Life Tweets (total) | NA | NA | NA | NA | 6148 |
| Real-Life Tweets (filtered) | NA | NA | NA | NA | 4163 |

* Tweets are not categorized by event type.

In Table 3, the ChatGPT-generated narratives and tweets from Table 2 are further filtered to exclude narratives or tweets that do not contain any PANAS *sentiment* keywords within any of the PANAS categories. For the Twitter set, 249 students were recruited from the Research Experience Program in the Psychology departments at Old Dominion University and Minnesota as part of the project [43]. Each student was required to have an active account with publicly available tweets. Active accounts were defined as following at least 30 other accounts, being followed back by at least 1/3 of those they follow, and having posted a comment at least once per month for the past 3 months. All tweets within a student's timeline from the previous year of their enrollment were collected and scored for sentiment analysis. The total number of tweets collected was 6148. Of the 6148 tweets, 4163 included terms that remained after being filtered out if the tweet was quoted, was a retweet, or was not in English. Of the 4163 tweets, 944 remained after checking for the presence of any PANAS sentiment. The tweet set serves as a baseline for comparing sentiment analysis results between the human tweets and ChatGPT-generated narratives. It was collected and used in this analysis because it is a dataset that is representative of active users (as opposed to automated accounts or bots) over a substantial period of time (1 year) engaging in discussions on a broad range of topics (as opposed to a single hashtag or categorized life event).

**Table 3.** Number of ChatGPT-generated narratives and real tweets containing any PANAS lexicon subgroup word, as described in column 6 of Table A1, by life event type.

| Event Type | Num. PANAS Sentiment ChatGPT Narratives | Num. PANAS Sentiment Tweets |
|---|---|---|
| Birth | 5586 | 944 * |
| Death | 5115 | 944 * |
| Hiring | 5447 | 944 * |
| Firing | 5447 | 944 * |

* Tweets are not categorized by event type.

## 3. Results

### 3.1. LLM Structured Prompt for Narrative Generation

We develop an LLM narrative prompt design as a structured prompt for shaping simulation data alongside LLM-specific directions in order to generate realistic narratives that can reflect the emotional, social, and cognitive states of simulated agents over time as well as accounting for relationships between agents. Our exploration utilized ChatGPT with the GPT-3.5 LLM; as such, our structured prompt is specific to GPT-3.5, though it might work similarly with other LLMs. Promisingly, GPT-3 has shown overall performance improvements when given less versus more prompting on how to perform a task [87]. Structural mechanisms that are stable, consistent, and precise are beneficial when attempting to generate messaging that realistically varies based on the context of the setting [4]. Therefore, we set about an iterative process of developing a structured input that could be utilized to provide consistent, transparent, and reproducible requests to an LLM platform to help better frame factual (based on the provided input information), on-topic, relevant, and time-frame-appropriate responses.

Our structured prompt defines a consistent, transparent, and reproducible method for providing a prompt to ChatGPT to generate the desired agent narrative. Preliminary testing using only descriptive-text inputs confirmed that ChatGPT is sufficiently capable of generating narratives that correctly describe a defined scenario. As a result, we transitioned to using a structured prompt with enumerated fields much less reliant on large amounts of descriptive text within the inputs. Section 3.1.1 documents our experiments and provides example inputs and outputs utilized throughout the evolution of the LLM Narrative Prompt Structure, and Section 3.1.2 defines the final version of the structured prompt.

#### 3.1.1. Experiments with Preliminary Designs

The first iteration of the structured prompt allowed for verbose text entries in its twelve fields and included a textual description of the *situation*, as shown in Figure 3a. This information was generated organically to test ChatGPT's capabilities and not based on simulated information. The narratives provided in Figure 3b demonstrate satisfactory outcomes; however, the structured input format was not practical for our purposes as it would have placed a burden of generating descriptive prose on the ABM which would have defeated the purpose of using an LLM chatbot.

In the second iteration, some descriptive text has been replaced with ABM simulated data and additional fields (e.g., `Voice`, `Narrative Immediacy`, and `Intended Emotion Level`) were added to the prompt structure, as shown in Figure 4a. The prompt still required the manual creation of more text than was desired. We also identified that a lack of clarity regarding the identity of the narrator, the subject of the narrative, and the nature of the subject being described was hindering the narrative generation process. This problem is demonstrated in Figure 4b where the prompt was incorrectly interpreted and Samantha is attributed the role of *newborn child* instead of *mother*. The figure also demonstrates an instance of unwanted messaging generated both before and after the requested narratives.

Situation: Paul (male, 42) and Ally (female, 36) Greenberg have just given birth to a baby boy, Max. It is their first child. Paul and Ally have been married for several years.

Identified Narrative Components:
(1) Agent Characteristics: the agent is white, male, 42 years old; he has never been married and has no children; he has a B.S. degree in marketing; he is left-leaning politically; he owns a home, lives in a small city, and is financially comfortable
(2) Connection: the target of the discussion is a group of other agents, the new parents
(3) Time: this life event is happening now
(4) Target of Narrative: the target audience is the parents and acquaintances of the parents
(5) Emotional/Sentimental Spectrum: the emotion of the agent can range from mildly to extremely positive
(6) Urgency: the agent is active
(7) Object of Narrative: there is a key life event, a birth
(8) History: the agent went to college with the new father; they were roommates; they have stayed in touch over the years and now live a couple hours away from each other
(9) Narrative Response Volume: generate 10 different narratives (tweets)
(10) Length of Narrative: each narrative should be 280 characters or less
(11) Special Tokens: hashtags and @s are allowed and encouraged
(12) Hyperlinks: hyperlinks are allowed

Generate a number of narratives (tweets) based on the defined Situation and 12 Identified Narrative Components.

(**a**)

1. Congratulations to my dear friends Paul and Ally on the arrival of baby Max! Wishing you all a lifetime of love, joy, and precious moments together. #NewParents #Blessed
2. The Greenberg family just got bigger and better! Welcome to the world, baby Max. Paul and Ally, you're embarking on the most incredible journey of parenthood. Cherish every second! 👶 #ParenthoodJoy #BabyMax
3. Introducing the newest member of the Greenberg clan, Max! Sending heartfelt congratulations to Paul and Ally on this beautiful blessing. May your lives be filled with endless love and laughter. 🎉 #NewBeginnings #BabyLove
4. It's a boy! Huge congratulations to my college roommate Paul and his amazing wife Ally on the arrival of their precious bundle of joy, Max. So thrilled for you both! 👶 #BabyBoy #ProudParents
5. Parenthood has officially begun for Paul and Ally! Baby Max, you've got the best parents in the world. Wishing you a lifetime of happiness, health, and incredible adventures together. #NewChapter #FamilyLove
6. The Greenberg family just got a little more adorable with the arrival of baby Max! Congratulations, Paul and Ally. Parenthood suits you both perfectly. Enjoy this incredible journey ahead! 👨‍👩‍👦 #BabyLove #NewParents
7. A warm welcome to baby Max, the newest addition to the Greenberg family! Sending heartfelt congratulations to Paul and Ally on becoming parents. Your lives will never be the same—in the best possible way! 👶 #BundleOfJoy #FamilyHappiness
8. Congratulations to Paul and Ally on the birth of their son, Max! Your lives are forever changed and filled with a love like no other. Wishing you endless moments of wonder and bliss as you embrace this incredible new chapter. #NewbornJoy #ProudParents
9. It's a beautiful baby boy! Sending my warmest congratulations to Paul and Ally on the arrival of Max. May your days be filled with cuddles, laughter, and the purest form of love. Enjoy this magical journey of parenthood! 🎉👶 #NewAdventure #BabyMax
10. The world just gained a little champion named Max! Congratulations to my dear friends Paul and Ally on the birth of their baby boy. This incredible journey will be filled with endless love, laughter, and priceless memories. #BabyMax #ParenthoodBliss

(**b**)

**Figure 3.** First-Iteration Structured-Prompt Input and ChatGPT Output. (**a**) Initial set of instructions tested with ChatGPT for generating narratives based on ABM Agents' life events and information. (**b**) Sample output set based on initial instructions presented in (**a**). Not all emojis have rendered.

Identified Narrative Components:

(1) Narrative Event Type: birth

(2) Event-Agent Name: Samantha

(3) Narrating Agent Relationship To Event Agent: daughter

(4) Narrating Agent Characteristics: location, Lambeth_Neighbourhood, age, 40, current education, 16, total eduation, 16, employer, employment_Places[2]( Current_Employed = 245, Number_Of_Jobs = 958, Minority_Friendly = 0.6294265811170259 ), employment, EMPLOYED, gender, MALE, generation, 1, group, MAJORITY_GROUP, income, 346276.0185210179, marital status, MARRIED, religion, MAJORITY,

(5) Narrative Event Characteristics: assume some plausible scenario around the life event type, with the event agent as the protagonist in the scenario; assume a set of characteristics about the event agent with the same fields as that of the narrating agent, the values of which can be similar to or different from those of the narrating agent

(6) Event Timeframe: this life event is happening now

(7) Target Audience: the target audience of the generated narrative is the event agent and shared followers

(8) Voice: yes, use an active voice to generate the narrative

(9) Narrative Immediacy: yes, use narrative immediacy to generate the narrative

(10) Intended Emotion Level: use emotion level 0.865 in the continuous emotional range [-1,1] to generate the narrative, where -1 is strongly negative, 0 is neutral, and 1 is strongly positive

(11) Event Agent History: past life events involving the event agent are defined in the following Pandas DataFrame output; use the information from these past life events to more fully develop the event agent and create a more complete and more descriptive narrative of the current life event

Event Type Event-Agent ID Event-Agent Name Event-Origin Time (weeks)... Group Income Marital Status Religion
98329 birth 1377 Samantha 0.011 ... MAJORITY_GROUP 346276.018521 MARRIED MAJORITY

[1 rows x 18 columns]

(12) Number of Narratives: generate 10 different narratives (tweets)

(13) Maximum Length: the generated narrative cannot exceed the maximum length of a tweet, 280 characters

(14) Special Tokens: yes, emojis, hashtags, and @s in the generated narrative are allowed and encouraged

(15) Hyperlinks: yes, hyperlinks, real or imaginary, may be included in the generated narrative

Generate a number of narratives (tweets) based on the 15 Identified Narrative Components.

(**a**)

Certainly! Based on the identified narrative components, here are 10 narratives (tweets) about the birth of Samantha:

1. "🎉 It's a moment of pure joy and wonder as Samantha enters the world, bringing love and happiness to our lives. Welcome, little one!  #Newborn #Blessed"

2. "Introducing Samantha, the newest member of our family. Her arrival has filled our hearts with immense love and excitement. The journey of parenthood begins! 👶 #BabyGirl #Miracle"

3. "Today, we celebrate the birth of Samantha, a precious gift who has already brightened our lives with her presence. We are grateful for this beautiful blessing.  #NewBeginnings #FamilyLove"

4. "Thrilled to announce the arrival of Samantha, our bundle of joy. She's a symbol of hope and a new chapter in our lives. Sending her all the love in the world! 🌈 #BabyLove #ProudParents"

5. "Meet Samantha, a little miracle who has made our family complete. Her arrival has filled our home with laughter and endless moments of happiness. We're so grateful! 👶❤️ #NewFamilyMember #CherishedBlessing"

6. "Welcoming Samantha, our little angel, into the world. Her presence is a reminder of the beauty and magic of life. We're excited to embark on this incredible journey together!  #NewbornJoy #FamilyBond"

7. "Announcing the birth of Samantha, our precious princess. She's already captured our hearts with her innocence and charm. Life has taken on a whole new meaning. 👑 #BabyGirl #BlessedMoments"

8. "We're overjoyed to share the news of Samantha's birth. She's a reflection of our love and a dream come true. Our hearts are overflowing with happiness and gratitude.  #LittleMiracle #JoyfulParents"

9. "With great joy, we introduce Samantha to the world. She's a precious soul who brings light and warmth wherever she goes. May her life be filled with love and endless possibilities!  #WelcomeBaby #NewAdventure"

10. "Celebrating the birth of Samantha, a true blessing in our lives. Each day with her is a treasured moment, reminding us of the miracles that surround us. We're forever grateful. 🌈✨ #CherishedGift #FamilyLove"

Congratulations to Samantha and her family on this special occasion! 🎉👶"

(**b**)

**Figure 4.** Second-iteration structured prompt input and a sample of ChatGPT's output. (**a**) Second-iteration design of prompt instructions tested with ChatGPT for generating narratives based on ABM agents' life events and information. (**b**) Sample output set based on instructions presented in (**a**) for second structured prompt iteration. Not all emojis have rendered.

For the third iteration, a long trial-and-error and fine-tuning interaction took place using the web-accessible version of ChatGPT. During this process, we tested the inclusion of additional fields, reducing the number of fields, and the renaming of existing fields to explore the impact on the generated narratives. Too little direction commonly resulted in wrong responses as a result of ChatGPT:

- Adding to or embellishing narratives with information that was not provided and that it had no basis for knowing;
- Creating temporal associations to information from the past which was not provided (i.e., relating present information based on unknowable changes from the past);
- Including congratulatory messaging beyond the scope of the requested narrative content.

This finding was expected and holds with related findings that ChatGPT requires fine-tuning for question-answering tasks [40] and directed prompting for decision support [41]. The final form of the structured narrative prompt is provided in the following section and satisfactorily addressed the issues identified during the first two iterations.

3.1.2. Final LLM Narrative Prompt Structure

A primary challenge faced was generating the right point of view for the narration and properly conveying the relationship between the narrator and the subject of the narrative. Narratives generated for cases where the narrator was reflecting its own information commonly produced reasonable results. However, narratives where the narrator was describing a life event that did not originate from the narrator, such as a discussing the birth of a neighbor's child, were less likely to properly convey the relationship between narrator and subject. As a result, we expanded the initial 12 fields into a structured prompt comprising 17 fields.

The first 11 fields of the structured prompts pertain to the content and context of the narrative being generated. This includes the life event type driving the narrative, the subject of the narrative, the subject's characteristics and relationship to the narrator (the relationship can be self-targeting), the narrator and the narrator's characteristics, the tense and voice of the narrative, a targeted sentiment level to help control the emotional content of the narration, and whether the narrative should be conveyed using narrative immediacy. Narrative immediacy helps in reflecting the viewpoint of the agent and providing a more engaging and intense messaging. Fields 12-16 provide context-independent direction to the LLM platform. This includes instructions such as how many narratives to generate, the maximum length of the narrative (this allows for boundaries to be set based on any intended outlets for disseminating the narratives), and whether special tokens or hyperlinks should be used within the narratives. The final field provides instructions for how the LLM platform should interpret the list. The following numbered list provides the 17 fields, along with their descriptions, that comprise the final form of the structured narrative prompt.

1. **Narrative Event Type**: What is the life event for which a narrative is being created? This can be anything deemed relevant for an agent such as a birth, marriage, change in education, etc.
2. **Subject of Narrative**: the agent, person, etc. that is the focus of the narrative.
3. **Subject's Relationship to Narrator**: What is the relationship between the narrator and the subject? Is the narrator referring to itself, a family member, a friend, a co-worker, a romantic connection, etc.?
4. **Subject's Characteristics**: a set of characteristics pertaining to the subject that are relevant to the narrative event.
5. **Narrator's Characteristics**: a set of characteristics that are relevant for the creation of the narrative with respect to the narrator, such as age or gender.
6. **Narrative Tense**: past, present, or future.
7. **Target Audience**: Who are the intended readers of the narrative and/or what is the intended medium of the narrative, such as Twitter, email, text message, diary, etc.?
8. **Voice**: Should the narrative use active or passive voice?
9. **Narrative Immediacy**: Should the narrative be conveyed using immediacy? Immediacy provides a more intimate, generally first-person, connection between the narrative and the reader.
10. **Maximum Temporal Proximity**: In the narration, how much time has passed since the life event occurred?

11. **Target Sentiment Value**: the intended level of emotion to convey in the narrative from −1 to +1 with −1 being strongly negative, 0 being neutral, and +1 being strongly positive.
12. **Subject's History**: the set of historical life events that support or expand upon the current narrative event, if any, such as prior Birth events when narrating a new birth.
13. **Number of Narratives**: the number of narratives to generate using the above criteria.
14. **Maximum Length**: the maximum length and unit of measure of the narrative being generated, i.e., characters, words, tokens, etc.
15. **Special Tokens**: Should the narrative include special tokens such as hashtags and @'s?
16. **Hyperlinks**: a set of hyperlinks to include in the narrative, if any.
17. **Instructions**: set of instructions for getting the large language model to understand how to use this list of criteria as well as any additional instructions needed to hone focus onto only the desired narrative materials.

ChatGPT was able to transform a rigidly defined structured prompt, devoid of any stem narrative, into a compelling, sentiment-driven narrative, as shown in Figure 5. Compared with Figure 4, the narrator and subject relationships are now defined appropriately for use by ChatGPT. The inclusion of field 17 in Figure 5a was necessary to achieve a proper relationship to the subject of the narrative and for preventing the creation of unrequested additional text. However, ChatGPT failed to observe the `Maximum Temporal Proximity` constraint, which was found to be a frequent problem when requesting multiple narratives in one response. There is more discussion on this issue and a proposed solution in Section 4.1.

ChatGPT utilizes a continuous $[-1, 1]$ scale for sentiment analysis, where −1 is associated with strongly negative sentiment, +1 is associated with strongly positive sentiment, and 0 is neutral. We manually tested for observable differences between narratives using almost identical prompts but with different target sentiment values. Some examples follow:

- Target sentiment value, −1: "Just got off the phone with Joyce, my unemployed mom. She's been looking for work for over two years now. It's disheartening to see her struggle. #Unemployment #JobSearch";
- Target sentiment value, +1: "It's been 104 weeks since my mom, Joyce, faced unemployment. She's a fighter and won't give up. Let's cheer her on! 💪 #Unemployment #Resilience".

For testing using narrative immediacy, ChatGPT defined narrative immediacy as a sensation of "being present or experiencing life events in real-time within a narrative", through the reading of which the audience has "a heightened emotional and sensory experience", which is "achieved through various techniques, such as using present tense, vivid descriptions, sensory details, and employing a first-person or close third-person perspective". Again, we manually tested for observable differences. In this case, we prompted for Birth event narratives with and without narrative immediacy. Some examples follow:

- With narrative immediacy: "The world welcomes Baby Max, the precious son of Paul and Ally. May he grow up surrounded by love, joy, and all the wonders life has to offer. Congratulations on this incredible blessing, and may your family be filled with happiness. #BabyBoy #NewestMember";
- Without narrative immediacy: "Sending my heartfelt congratulations to Paul and Ally on the birth of their precious baby boy, Max! May this new journey be filled with endless love, joy, and beautiful memories. #NewParents #BabyMax".

Identified Narrative Components:
(1) Event: has a new baby
(2) Subject of Narrative: Samantha
(3) Subject's Relationship to Narrator: daughter
(4) Subject's Characteristics: Location,Ealing
(5) Narrator's Characteristics: Age,48, Employer,employment_Places[4]( Current_Employed = 186, Number_Of_Jobs = 190, Minority_Friendly = 0.3944534424065025 ), Employment,EMPLOYED, Gender,MALE, Generation,1, Group,MAJORITY_GROUP, Income,484371.17162391735, Marital Status,MARRIED
(6) Narrating Tense: present
(7) Target Audience: Twitter, all audiences
(8) Voice: active
(9) Narrative Immediacy: yes
(10) Maximum Temporal Proximity: 24 hours
(11) Target Sentiment Value: 0.17
(12) Subject's History: none
(13) Number of Narratives: 10
(14) Maximum Length: 280 characters
(15) Special Tokens: yes
(16) Hyperlinks: yes
(17) Instructions: Generate a numbered set of narratives (tweets) based on the previous 16 Identified Narrative Components (INCs). Narratives are from the perspective of the narrator whose characteristics are defined in INC(5) and should be age-appropriate, given the narrator's age defined in INC(5). The subject of the narrative, named in INC(2), performs the life event in INC(1). A relationship of "self" in INC(3) indicates the narrator and the subject are the same person. Each narrative must have a temporal relationship with the life event that is constrained by the maximum temporal proximity defined in INC(10). Do not add any text or special tokens outside of the numbered set of generated narratives.

(**a**)

1. My daughter Samantha just had a new baby! I'm so excited to welcome my grandchild into the world. #newbaby #grandparent #familylove
2. Samantha and the baby are doing well. I can't wait to visit them in Ealing and hold my grandchild for the first time. #familyvisit #newborn #grandparent
3. It's been a few days since the baby was born and I'm still over the moon. Being a grandparent is such a special feeling. #blessed #familylove #newgrandparent
4. I've been helping Samantha with the baby and it's been a joy to watch her become a mother. #proudmom #newmom #familybonding
5. I can't believe how quickly the baby is growing. It feels like just yesterday Samantha was announcing the pregnancy. #timeflies #newborn #familymemories
6. Samantha and I have been reminiscing about when she was a baby. It's amazing to see her now as a mother herself. #familyhistory #newmom #familybonding
7. I'm so grateful for this new addition to our family. The baby has brought so much joy and love into our lives. #familylove #newbaby #blessed
8. Samantha and her partner are doing an amazing job with the baby. I'm proud of the parents they've become. #newparents #familypride #parenting
9. I can't wait to see what the future holds for my grandchild. I know they'll be surrounded by love and support from our family. #familyfuture #newgeneration #familylove
10. As I hold my grandchild, I'm reminded of the preciousness of life and the importance of family. I'm grateful for this moment and for my daughter Samantha. #familybonding #newbaby #grateful

(**b**)

**Figure 5.** Final-design structured prompt input and ChatGPT output. (**a**) Final-prompt-design set of instructions tested with ChatGPT for generating narratives based on ABM agents' life events and information. (**b**) Sample output set based on initial instructions presented in (**a**).

### 3.2. Statistically Significant and Not Statistically Significant Differences

The comparison of tweets and ChatGPT-generated narratives used the 11 PANAS traits and four life event types combined to produce a total of 44 tested comparisons for significant differences. As outlined in Section 2.1, Fisher's exact test is applied in cases where two-by-two contingency tables contain any cells with fewer than five samples. Chi-squared tests are applied in all other cases. The two-by-two contingency tables are constructed with tweets and ChatGPT-generated narratives as columns and the counts of

0s and 1s with respect to a PANAS trait as the two rows. As such, 11 contingency tables are constructed with one specific to each PANAS trait. For all of our tested cases, all cells within the contingency tables resulted in greater than five samples; therefore, the chi-squared test was applied for all significance tests.

Statistical significance tests are applied using the mean sentiment scores for a PANAS trait between tweets and ChatGPT-generated narratives. For each narrative, as well as for each tweet, the PANAS trait assessments are binary. A score of 1 is assigned for a trait if any corresponding words within the lexicon are contained within the text and a score of 0 is assigned if no terms are contained. As a result, the mean value for each assessment reflects the prevalence of the corresponding sentiment within each dataset.

Of the 44 tests for statistically significant differences between the tweets and the ChatGPT-generated narratives for the Birth, Death, Hired, and Fired event types, four comparisons provided evidence in support of rejecting the null hypothesis in favor of the alternative hypothesis and 40 comparisons provided evidence in support of failing to reject the null hypothesis. Table 4 provides the aggregate number of comparisons for rejecting or failing to reject the null hypothesis for each life event type.

**Table 4.** Number of comparisons supporting rejecting or failing to reject the null hypothesis.

| Life Event | Number of Comparisons for Rejecting the Null Hypothesis | Number of Comparisons for Failing to Reject the Null Hypothesis |
|---|---|---|
| Birth | 10 | 1 |
| Death | 10 | 1 |
| Hired | 11 | 0 |
| Fired | 9 | 2 |

We are particularly interested in instances where no statistically significant evidence is found in support of rejecting the null hypotheses outlined in Hypothesis 0. These instances represent cases where the sentiment prevalence for a given PANAS group was not discernibly different from the language used within real tweets and represent a step forward in the generation of realistic narratives from simulated agents. For the ChatGPT-generated narratives, four of the forty-four comparisons (9.09%) were not statistically significantly different from the sentiment prevalence within the tweets. For Birth events, this included *all negative* (ChatGPT-generated narratives mean = 0.410, tweets mean = 0.438, *p*-value = 0.128). For Death events, this included *nervous_jittery* (ChatGPT-generated narratives mean = 0.052, tweets mean = 0.074, *p*-value = 0.052). For Fired events, this included *all positive* (ChatGPT-generated narratives mean = 0.506, tweets mean = 0.514, *p*-value = 0.691) and *strong_active* (ChatGPT-generated narratives mean = 0.322, tweets mean = 0.319, *p*-value = 0.859). No PANAS group comparisons were found to be not statistically significantly different for Hired comparisons.

Mean binary PANAS scores for all sentiment keyword categories and all life event types are plotted in Figure 6. For *Death* and *Fired* narratives, it was expected that sentiments would skew towards the use of negative terms. These narratives did generally attain higher mean sentiment scores compared to the mean scores for *Birth* and *Hired* narratives; however, *Death* narratives still achieved higher mean sentiment in the positive polarity categories of *excited_enthusiastic_inspired*, *proud_determined*, and *strong_active*. Additionally, the ChatGPT-generated Birth narratives unexpectedly attained the highest mean sentiment in the negative polarity category of *nervous_jittery*. For the six negative PANAS categories, the mean tweet sentiment scores fell in the middle of all other event types, excepting *nervous_jittery*, where it scored the lowest mean sentiment value of all narrative sources.

Across all positive categories, the tweets are much less positive than ChatGPT-generated Birth, Death, and Hired narratives. For *Birth* and *Hired* narratives, it was expected that mean sentiments would skew higher for positive PANAS groupings. *Birth* narratives scored the highest mean sentiment in all five positive polarity groupings. *Birth* and *Hired* narratives achieved higher mean sentiment scores than the tweets in all positive categories. However,

ChatGPT-generated Death narratives scored the second-highest mean sentiment in all positive categories aside from *interested_attentive_alert*. Mean sentiment scores for ChatGPT-generated Fired narratives and for the tweets are the lowest-scoring narrative sources for all positive PANAS polarity groupings. The mean sentiment of the ChatGPT-generated Hired narratives are at or near the middle ranking for every positive category.

Comparing the ChatGPT-generated Birth narratives against the tweets, the mean binary prevalence of PANAS category terms differed at a statistically significant level for 10 PANAS categories, as shown in Figure 7. This includes five positive and five negative categories. The only category for which they did not differ significantly is *ANY negative* (ChatGPT-generated narratives mean = 0.410, tweets mean = 0.438, *p*-value = 0.128). Mean tweet sentiment scores were higher in four of the six negative categories, while ChatGPT-generated narratives scored higher for *hostile_irritable* (ChatGPT-generated narratives mean = 0.362, tweets mean = 0.325, *p*-value = $3.37 \times 10^{-2}$) and *nervous_jittery* (ChatGPT-generated narratives mean = 0.334, tweets mean = 0.074, *p*-value = $1.99 \times 10^{-58}$) categories. ChatGPT-generated narratives scored much higher in all five positive categories, with mean values exceeding the upper standard deviation values of the tweets for *excited_enthusiastic_inspired*. Larger magnitudes in the differences of mean sentiment scores between ChatGPT-generated Birth narratives and the tweets are observed for all of the positive PANAS categories compared to all of the negative PANAS categories. All computed values for comparing ChatGPT-generated narratives and the tweets are provided in Table A2.

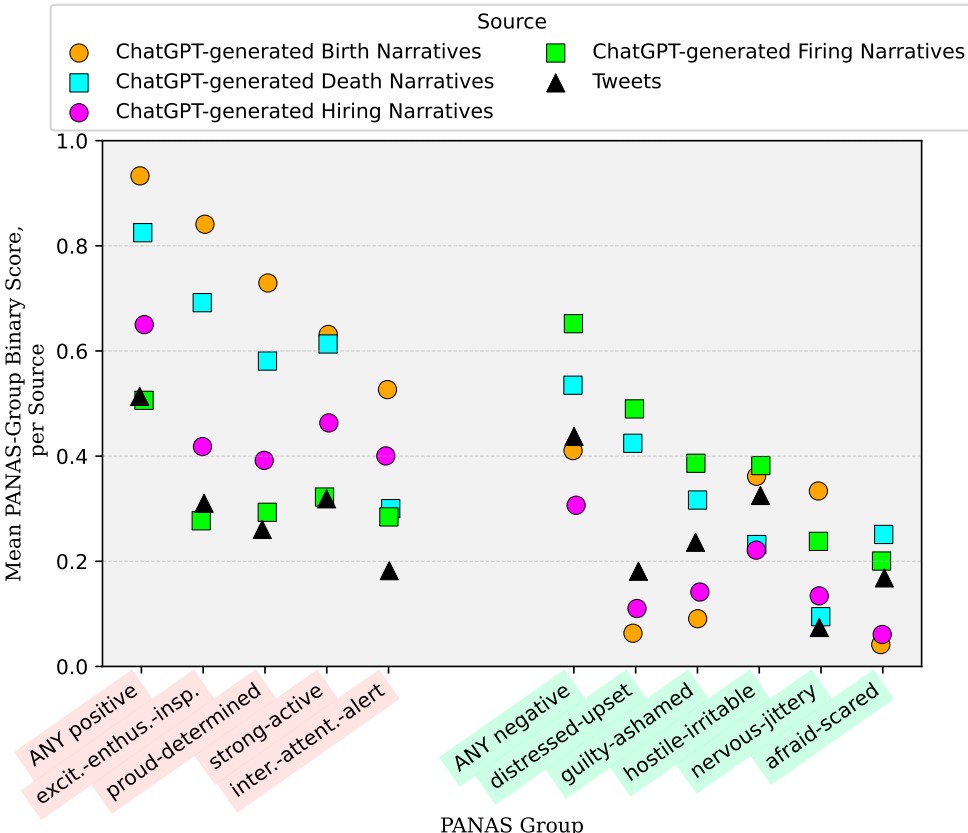

**Figure 6.** PANAS category mean binary scores for ChatGPT-generated narratives per life event type and for all tweets. Positive keyword labels and expected-positive event markers have warm colors; negative keyword labels and expected-negative event markers have cool colors.

Comparing the mean sentiment of PANAS groups between ChatGPT-generated *Death* narratives and tweets yielded statistically significantly different mean scores for 10 of the PANAS categories, as shown in Figure 8. This includes five positive and five negative

categories. The only category for which they did not statistically significantly differ is *nervous_jittery*. ChatGPT-generated *Death* narratives scored higher in five of the six negative categories, while tweets outscored ChatGPT-generated narratives in *hostile_irritable* (ChatGPT-generated narratives mean = 0.232, tweets mean = 0.325, *p*-value = $1.25 \times 10^{-9}$). ChatGPT-generated narratives scored higher in all five positive categories, as well as for *ANY positive* (ChatGPT-generated narratives mean = 0.825, tweets mean = 0.514, *p*-value = $2.27 \times 10^{-98}$). Greater differences between ChatGPT-generated *Death* narratives and tweets are observed on average for positive PANAS categories. All computed values for comparing ChatGPT-generated narratives and the tweets are provided in Table A3.

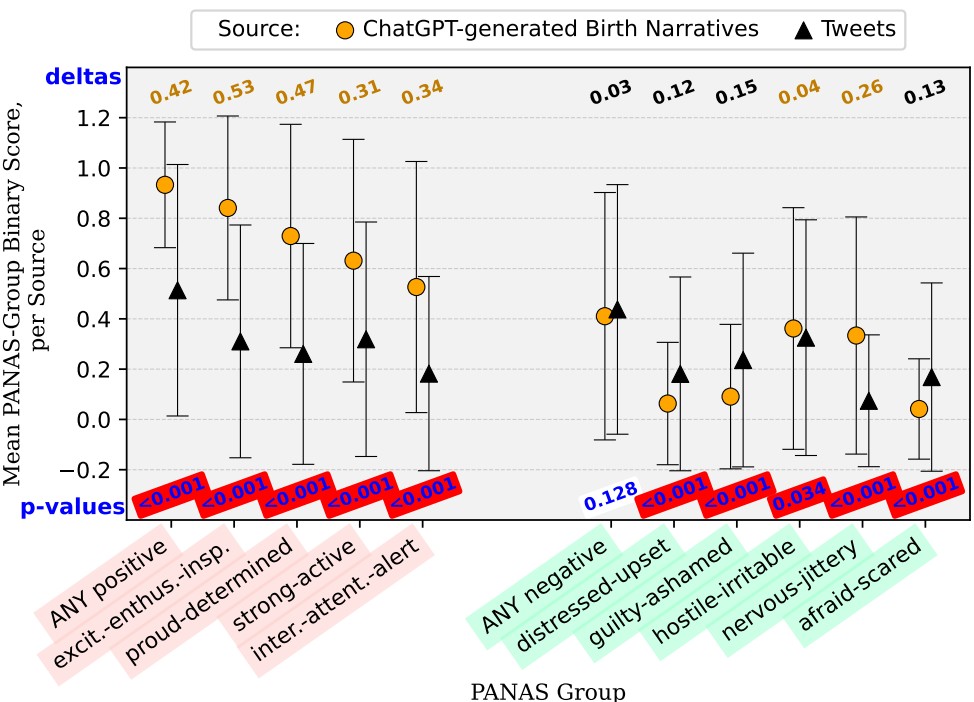

**Figure 7.** Mean and standard deviation sentiment scores for ChatGPT-generated *Birth* narratives versus tweets. The *p*-values corresponding to the chi-Squared test for each PANAS group pairing is displayed along the bottom. *p*-values less than $\alpha = 0.05$ (red) indicate a statistically significant difference in mean binary sentiment value. The deltas convey the difference between mean values and are colored according to the higher source.

Comparing ChatGPT-generated *Hired* narratives and tweets, the mean binary prevalence of PANAS groups is statistically significantly different for all 11 categories, as shown in Figure 9. The tweets scored higher in five of the six negative categories, while ChatGPT-generated narratives outscored the sentiment scores of the tweets in *nervous_jittery* (ChatGPT-generated narratives mean = 0.134, tweets mean = 0.074, *p*-value = $3.55 \times 10^{-7}$). ChatGPT-generated narratives scored higher in all five positive categories, including *ANY positive* (ChatGPT-generated narratives mean = 0.650, tweets mean = 0.514, *p*-value = $1.57 \times 10^{-15}$). Slightly greater differences between ChatGPT-generated *Hired* narratives and tweets are observed on average for all positive PANAS categories. All computed values for comparing ChatGPT-generated narratives and the tweets are provided in Table A4.

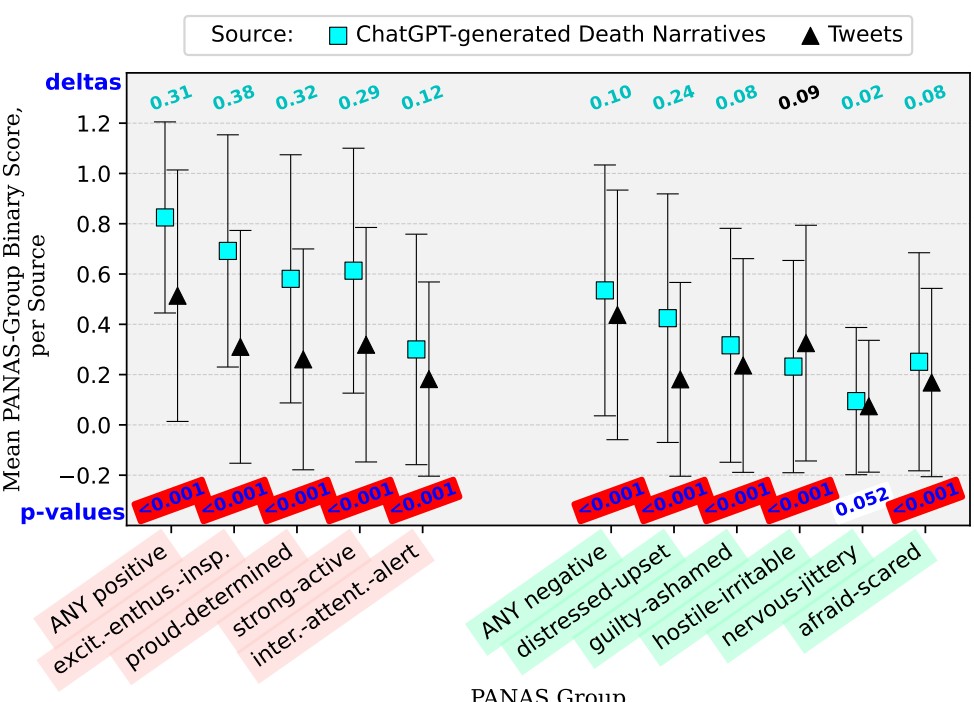

**Figure 8.** Mean and standard deviation sentiment scores for ChatGPT-generated *Death* narratives versus tweets. The *p*-values corresponding to the chi-Squared test for each PANAS group pairing are displayed along the bottom. *p*-values less than $\alpha = 0.05$ (red) indicate a statistically significant difference in mean binary sentiment value. The deltas convey the difference between mean values and are colored according to the higher source.

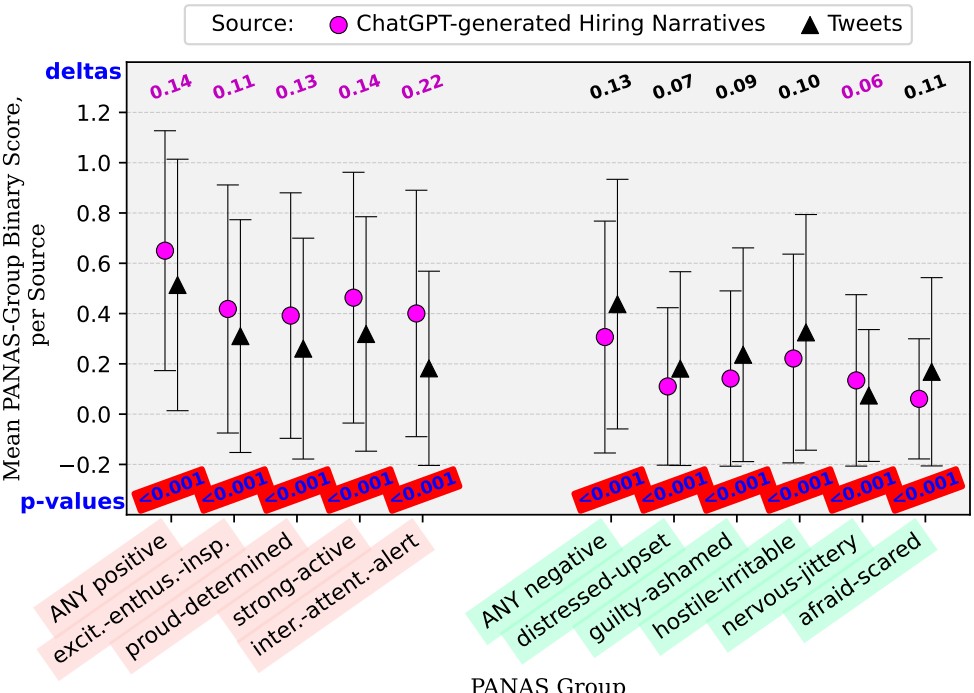

**Figure 9.** Mean and standard deviation sentiment scores for ChatGPT-generated *Hired* narratives versus tweets. The *p*-values corresponding to the chi-Squared test for each PANAS group pairing are displayed along the bottom. *p*-values less than $\alpha = 0.05$ (red) indicate a statistically significant difference in mean binary sentiment value. The deltas convey the difference between mean values and are colored according to the higher source.

Comparing ChatGPT-generated *Fired* narratives and Twitter narratives, the mean binary prevalence of PANAS category terms differed at a statistically significant level for nine categories, as shown in Figure 10. This includes three positive and six negative categories. The only categories for which they do not differ significantly are *ANY positive* and *strong_active*. ChatGPT scored higher in all six negative categories. The tweets scored higher in two positive categories, including *ANY positive* and *excited_enthusiastic_inspired* (ChatGPT-generated narratives mean = 0.277, tweets mean = 0.310, *p*-value = $3.91 \times 10^{-2}$). Greater differences between ChatGPT-generated *Fired* narratives and tweets are observed on average for negative PANAS categories. This is the only event type for which the ChatGPT-generated narratives and the tweets exhibit greater difference for negative categories than positive categories. All computed values for comparing ChatGPT-generated narratives and the tweets are provided in Table A5.

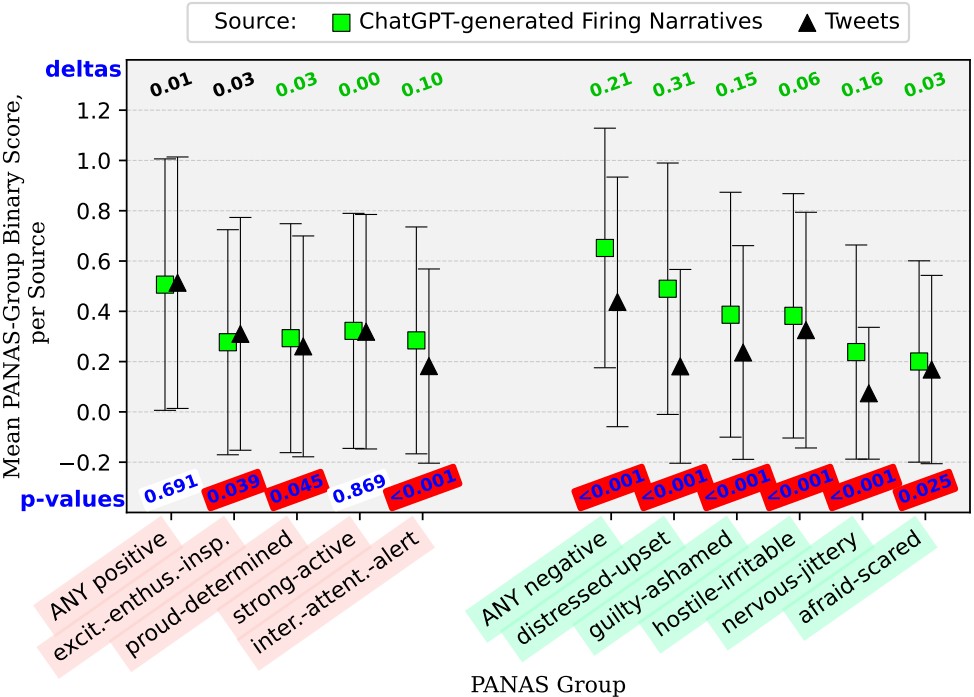

**Figure 10.** Mean and standard deviation sentiment scores for ChatGPT-generated *Fired* narratives versus tweets. The *p*-values corresponding to the chi-Squared test for each PANAS group pairing are displayed along the bottom. *p*-values less than $\alpha = 0.05$ (red) indicate a statistically significant difference in mean binary sentiment value. The deltas convey the difference between mean values and are colored according to the higher source.

## 4. Discussion

The provided LLM Narrative Prompt Structure provides a consistent, transparent, and reproducible process for requesting narratives generated by LLMs. This structure assumes that a event type is known, that there is a relationship between the desired narrator of the life event and the subject of the life event (even if this just involves specifying that the narrator is describing a life event that happened to him/her/itself), and that some amount of information is known about the life event with respect to the intended subject of the narrative. Our results support that the current form of the LLM Narrative Prompt Structure can produce realistic narratives based on sentiment term prevalence comparisons to real tweets based on the utilized PANAS trait lexicon.

In most cases, the average sentiment value of the ChatGPT-generated narratives was higher than the sentiment observed from the real tweets across all of the positive PANAS groups. The average sentiment of the tweets with respect to negative PANAS categories fell in the middle of the ChatGPT-generated narrative sentiments across the four life event types.

Overall, the magnitudes of the differences in mean sentiment scores for negative PANAS groups were smaller than the magnitudes for positive traits for Birth, Death, and Hired life event comparisons. However, negative groups showed higher magnitudes in differences for Firing comparisons. As such, additional work is needed to refine the structure for additional LLMs and for specific contexts and event types.

Overall, ChatGPT successfully generated narratives based on simulated agents' information using the structured narrative prompt. The resulting ChatGPT-generated narratives achieved sentiment levels that were statistically indiscernible from those of real tweets for four out of forty-four tested comparisons. The developed prompt structure provides consistency and clarity to the narrative generation process while making the data connections to the narrative transparent through the implementation of the prompt. This provides a measure of traceability that can be used to assess the validity of developed messaging and/or narration by comparing the instructions of the prompt to a generated message. The flexibility of the type of information that can be sent through the prompt using simulation data can enhance the ability to connect model users with the contextual plights, wishes, desires, injustices, and happiness of simulated environments by facilitating a connection between the simulated agents' perspectives and the perspectives of the model users, such as policy makers, researchers, decision makers, etc. This structure can also aid ABM researchers in distilling large volumes of agent information from simulation runs into a natural language format that is more easily digestible. In this manner, simulated agents' interactions, key life events, communications, feelings, emotional states, preferences, and/or beliefs can be more readily and simply communicated to desired target audiences. In the following subsections, we discuss many of the critical lessons learned that we gained throughout this research endeavor, we discuss limitations of our study, and we outline avenues for future work.

### 4.1. Lessons Learned

1. **Using the ChatGPT API for generating multiple, independent narratives.** The ChatGPT API was not well suited to generating multiple instances of a requested narrative in a single response. There was a strong tendency to narrate a continuous, temporally advancing story instead of a set of independent narratives describing a single life event. Using the `n` parameter in the Python API `ChatCompletion` function call appears to remedy this behavior, as ChatGPT generates a set of $n$ independent, unconnected responses. ChatGPT does not appear to retain knowledge of narratives 1 through $i$ when generating the $i$th narrative, when using the `n` parameter.

2. **Balancing creativity with correctness.** The level of stochasticity that ChatGPT employs for choosing the next token during text completion is moderated by the `temperature` parameter. A zero temperature outputs identical responses yielding repeated identical inputs. Increasing the temperature increases the set of next available tokens in the completion, effectively increasing the response space and allowing for greater variation among the responses. If the temperature is set too high and ChatGPT selects inappropriate tokens, this increased creative capacity can impact correctness. However, even when using a temperature of zero for the API, ChatGPT can still produce categorically incorrect responses, such as generating narratives about car fires and house fires when the prompt was to generate narratives about being fired from a job. We utilized a `temperature` of zero to (i) attempt to limit incorrect narratives and (ii) to address ChatGPT's tendency toward "storytelling", instead of generating multiple independent narratives in one response. This ultimately was not very effective, as noted in the previous lesson learned. Conversely, when using the `n` parameter to generate multiple, different narratives, the `temperature` value must not be zero as this results in only identical results. For this study, we assigned `n` to 1 as we requested multiple narratives to be provided in a single response. The default minimum and maximum `temperature` values for `ChatCompletion` API calls are 1, 0, and 2. The default value for `n` is 1.

3.　**ChatGPT API time-out errors.** The API fails frequently due to request time-out errors. Therefore, the experimental setup should account for this and should be able to resume efficiently after an error. For this study, a Python script reads prompt files from a directory and moves them to another directory after a successful response is received. In this process, if the script is restarted due to a time-out error, no prompts are lost or repeated.

*4.2. Limitations*

It is important to note some of the limitations of our study. The demographics, geographical location, and even previous daily activities of the authors of text-based data can result in substantially different word choices even when discussing the same subject matter. While it is beyond the scope of this work to control for these biases, it is important to note that they exist and could influence the analysis that informs this article. Geographic bias that influences the sentiment analysis of text-based social media messages has been highlighted in [88,89]. Another study shows that the sentiment of social media about a given subject can be biased by the time of day the message is authored and if the author is a resident or visitor in the city in which the message is composed [90]. Additionally, the reliability and validity of targeted narrative messaging, such as the narrator–subject relationship utilized in our prompt structure, needs further evaluation with specific attention to the context of the simulation setting to gauge how well the message serves as a true representation of the original life event.

1.　**Problem type.** For this study, ChatGPT was not required to solve complex problems or rely heavily on factual information from training data. All the required factual information, including narrator and subject characteristics, was provided in the prompt. ChatGPT appears well suited to this style of creative task and provides technically correct outputs in a majority of instances as long as the instructions and constraints in the input prompt are observed. Narratives can easily be validated manually by comparing prompt inputs to the created narratives. This differs from other types of tasks, such as asking ChatGPT to solve a mathematical problem or to diagnose a medical condition [28], which require domain knowledge of much more complex background information that is not included in the prompt. These problem types are much less subjective and not as easily validated. Further, ChatGPT currently cannot accurately provide sources or references for validating the response information. In this case, the human reader has to determine if the response is legitimate or if ChatGPT has "hallucinated" some trustworthy-sounding but incorrect response, without the benefit of reliable references [36].

2.　**Use case.** For this study, the generated narratives were not posted to social media or broadcast in any way, but were used solely for analysis. Incorrect narratives were identified manually and used to inform corrections to the structured narrative prompt but did not incur any other negative consequences. For use cases in which responses are not or cannot be validated by a human before utilization, there is a risk of dissemination of erroneous information. Numerous correct prior responses do not prevent incorrect responses from occurring in the future; in other words, there is no way to bound or know the response space [36]. As noted in the second lesson learned, even with minimal stochasticity, ChatGPT generated completely incorrect narratives about car fires and house fires, which could not have been predicted by the hundreds of preceding responses for that event type which did not do this.

3.　**ChatGPT API response speed.** As noted in the third lesson learned, the Python API regularly failed due to time-out errors, so this currently might not be an appropriate tool for situations with strict time constraints.

4.　**Token volume in real time/quicker than real time.** The ability to generalize our approach for real-time applications of ChatGPT for narrative generation is limited based on the token limit of the API. The current ChatGPT API version has a rate limit of 3500 requests per minute and 90,000 tokens per minute [91].

5. **Domain expertise.** The creation of LLMs is based on broad ranges of volumes of reference literature. It is important to determine that generated results are in line with the domain expertise of the targeted problem or system [92]. This article does not attempt to refine the learning base of the LLM for the narration of key life events, as the broad range of potential response types for individuals was desirable as the starting point for this effort. However, future avenues of research require assessing the validity of narratives within their respective domains, such as for births and Death events, and within a larger context, such as refugee camps, natural disaster response, etc.

6. **Underspecification hinders narrative generation.** Increased specificity in the prompting of desired narratives from ChatGPT has been beneficial in reducing the number of iterations for generating and assessing the correctness of the narratives. Similar to ML pipeline problems with underspecification, where underspecification in training leads to problems in reliability and validity [93], underspecification of narrative requests from ChatGPT led to many more erroneous responses and the expansion of the structure provided in this article.

7. **Tweet comparison sets.** Tweets were not categorized per event type like the generated narratives. The tweet set is utilized assuming that it represents a general sample of the population. As such, the generalizability of the sentiment findings should not be extended to other sample populations without proper supporting justifications about the reasonableness of the extension.

8. **Not Correcting for Multiple Comparisons.** For tests involving multiple comparisons, it is common practice to apply a correction to adjust the significance level of the applied hypothesis tests. However, no correction is advised if: (1) the study is restricted to a small number of planned comparisons; (2) the study is exploratory using post hoc testing of unplanned comparisons for further investigation; (3) multiple simple tests are envisaged and it is the results of the individual tests that are important; (4) it is imperative to avoid a type II error [94]. Meanwhile, a Bonferroni correction is advised if: (A) a single test of the universal null hypothesis that all tests are not significant is required; (B) it is imperative to avoid a type I error; (C) a large number of tests are carried out without preplanned hypotheses in an attempt to establish that any result may be significant [94]. The design of this study fits with all four points for when the correction is not necessary and it does not fit any of the three points pertaining to when to utilize the Bonferroni correction. Additionally, since we are interested in the individual comparisons for each PANAS category, not applying the Bonferroni correction is more appropriate for the design of this study as we are interested in identifying interesting patterns or potential relationships that may warrant further investigation. For this study, we are accepting a higher risk of Type I errors (false positives) to reduce the chance of Type II errors (false negatives). As such, this study is more willing to identify something as significant even if it might not be, in order to not miss any potential findings.

### 4.3. Future Work

Future work includes developing processes for verifying that the narratives generated by the LLM are correct based on the instructions provided in a more formal, consistent, and reproducible fashion. Similarly, a validation process is needed to assess that the narratives are valid given the perspective of an agent combined with the contextual information of the environment. A pipeline for transparently assessing whether each narrative within a set of narratives is correct or incorrect that can be extended to the automated assessment of the set of narratives as a whole would be beneficial to the LLM community. We intend to explore this process further as a follow-up article to this research.

Additional research is needed to expand the comparison of event-specific narratives with correspondingly categorized tweet datasets. This should help in yielding realism and contextual appropriateness for simulating specific types of narratives. This can also be

used as an avenue for developing contextually relevant perspectives from communities of interest, such as for displaced and marginalized communities. Research in this realm can also include determining how to better bridge the perspective of the simulated agent with the perspectives of potential policy and decision makers. Projects exploring socially relevant outcomes should engage stakeholders and domain experts from the start to the end of the design process. Reports on the biases and limitations of applying results should be produced to mitigate the potential for misuse and inappropriate application of results [92]. To facilitate trustworthiness in the responses generated by an AI-supported system, the system should provide fair, accurate, independent, honest, and reliable outcomes, among other values [95].

The capture and assessment of the temporal components of narratives is another avenue for continued research. Narrative generation can be expanded from single life events into a series of relevant and/or related life events. This could provide a more relevant narrative based on an agent's knowledge or history of past life events, i.e., current age, relationships, emotional state, etc. Combining pertinent history information with the social norms or cultural values of the society being modeled would help to connect with moral, ethical, and social underpinnings of the modeled system.

Further experiments with the design of the structured prompt for narrative generation may yield insight into the capabilities and limitations of LLM-based chatbots, like ChatGPT, to process structured prompts in varying levels of conversationality. That is, on one end of the spectrum, prompts are minimalistic and utilitarian, and on the other end, prompts are verbose and similar to conversational written or spoken language. These results could have implications beyond the domain of agent-based simulation, for example, in medicine, where it may become commonplace to submit diagnostic or treatment queries to an LLM, preferably in a concise, feature-based format [39]. Additionally, we want to expand the experimental LLMs in our study to include OpenAI's GPT-4 LLM, Meta's LLaMa or LLama 2 LLM, and Google's Bard chatbot with the PaLM 2 LLM.

## 5. Conclusions

This study provides a structured narrative prompt for consistently and transparently providing information to LLM platforms in order to yield narratives oriented around specific types of life events. Experimentation with the structured narrative prompt using OpenAI's ChatGPT-3.5 API showed that correct narratives can be generated with sentiment levels that are indiscernible from the sentiment of real tweets. This supports that the structured narrative prompt can serve as a promising avenue for generating sentimentally realistic narratives of simulated agents. Assessments for statistically significant differences in sentiment scores between ChatGPT-generated narratives and real tweets are generated using chi-Squared tests. Overall, four of the forty-four tested comparisons using Birth, Death, Hired, and Fired themed life events were found to be statistically indiscernible from real tweets. In 32 of the 44 comparisons, ChatGPT-generated narratives achieved higher average sentiment scores across the 11 PANAS groups. Additionally, the results clearly show that the sentiment levels conveyed alongside the ChatGPT-generated narratives varied widely across the four utilized event types.

Trends were observed tending towards higher mean sentiment of positive traits within the ChatGPT-generated narratives. Whether this tendency holds for other LLM platforms is not something that can be generalized from this study and research into the sentiment tendencies of other LLMs requires separate evaluation. Based on the iterative development process for developing the LLM structured narrative prompt, target sentiment level is already a primary input field (field 11); however, additional research is required to further calibrate the desired sentiment level for a given context or event type to potentially scale back the overall sentiment being generated within the created narratives.

Our study relates to desires expressed by the medical community for formulaic LLM prompts [39]. Our findings suggest that using a similar structured prompt can be helpful in obtaining useful LLM responses in critical settings such as healthcare, clinical training,

or risk communication. Our results are but a single data point in this emerging and quickly expanding field. However, researchers should conduct additional domain-specific research into the appropriateness of the LLM structured narrative prompt for other domains.

**Author Contributions:** Conceptualization, C.J.L., E.J.J., V.Z., K.O., E.F. and R.G.; methodology, C.J.L., E.J.J. and R.G.; software, C.J.L., R.G. and E.J.J.; validation, C.J.L., R.G. and E.J.J.; formal analysis, C.J.L., R.G. and E.J.J.; investigation, C.J.L., R.G. and E.J.J.; data curation, C.J.L., R.G. and E.J.J.; writing—original draft preparation, C.J.L., E.J.J. and R.G.; writing—review and editing, C.J.L., E.J.J., V.Z., K.O., E.F. and R.G.; visualization, C.J.L., E.J.J. and R.G.; funding acquisition, R.G. and C.J.L. All authors have read and agreed to the published version of the manuscript.

**Funding:** This research was supported, in part, by Old Dominion University through project 300916-010, Effective & Individualized Risk Communication [74].

**Institutional Review Board Statement:** Not applicable as this research did not involve the use of human subjects. However, the previously constructed Twitter dataset was approved as EXEMPT FROM IRB REVIEW by the Old Dominion University Institutional Review Board on 13 May 2022 per Exemption category #2 from federal regulations [43].

**Informed Consent Statement:** Not applicable.

**Data Availability Statement:** All the data, code, and generated narratives utilized in this article are freely accessible in an online repository [86].

**Conflicts of Interest:** The authors declare no conflict of interest.

## Abbreviations

The following abbreviations are used in this manuscript:

| | |
|------|-----------------------------------------|
| ABM | Agent-Based Model |
| API | Application Programming Interface |
| GPT | Generative Pre-trained Transformer |
| LLM | Large Language Model |
| PANAS | Positive and Negative Affect Schedule |

## Appendix A. PANAS Lexicon

*Breakdown of the PANAS Lexicon*

We utilize a PANAS scale [44] lexicon formed from a combination of the National Research Council (NRC), General Inquirer, and LWIC lexicons [70–72]. This lexicon provides the following 20 traits that can be assessed in text: *interested*, *alert*, *attentive*, *excited*, *enthusiastic*, *inspired*, *proud*, *determined*, *active*, *strong*, *depressed*, *upset*, *guilty*, *ashamed*, *hostile*, *irritable*, *nervous*, *jittery*, *afraid*, and *scared*. Table A1 provides the full hierarchical representation of the lexicon and denotes which traits are grouped together based on positive and negative polarities.

**Table A1.** PANAS Lexicon breakdown.

| PANAS_Trait | PANAS_Group | PANAS_Polarity | Lexicon | Lexicon_Subgroup | Lexicon_Subgroup_Description | Comments |
|---|---|---|---|---|---|---|
| interested | interested ǀ alert ǀ attentive | positive | NRC | anticipation | 839 words associated with anticipation through MTurk crowdsourcing | only include positive words |
| alert | interested ǀ alert ǀ attentive | positive | General Inquirer | Perceiv | 167 words associated with perception and perceiving | only include positive words |
| attentive | interested ǀ alert ǀ attentive | positive | NRC | anticipation | 839 words associated with anticipation through MTurk crowdsourcing | only include positive words |
| excited | excited ǀ enthusiastic ǀ inspired | positive | General Inquirer | Arousal | 67 words indicating excitation; aside from pleasures or pains; but including arousal of affiliation and hostility | only include positive words |
| enthusiastic | excited ǀ enthusiastic ǀ inspired | positive | General Inquirer | Arousal | 67 words indicating excitation; aside from pleasures or pains; but including arousal of affiliation and hostility | only include positive words |
| inspired | excited ǀ enthusiastic ǀ inspired | positive | NRC | joy | 689 words associated with joy through MTurk crowdsourcing | only include positive words |
| proud | proud ǀ determined | positive | NRC | trust | 1231 words associated with trust through MTurk crowdsourcing | only include positive words |
| determined | proud ǀ determined | positive | General Inquirer | Pleasur | 168 words indicating the enjoyment of a feeling. Including words indicating confidence; interest and commitment | only include positive words |
| active | strong ǀ active | positive | General Inquirer | Active | 1902 words implying strength | only include positive words |
| strong | strong ǀ active | positive | General Inquirer | Strong | 2045 words implying an active orientation | only include positive words |
| distressed | distressed ǀ upset | negative | General Inquirer | Pain | 254 words indicating suffering; lack of confidence; or commitment | only include negative words |
| upset | distressed ǀ upset | negative | NRC | sadness | 1191 words associated with sadness through MTurk crowdsourcing | only include negative words |
| guilty | guilty ǀ ashamed | negative | General Inquirer | Vice | 685 words indicating an assessment of moral disapproval or misfortune | only include negative words |
| ashamed | guilty ǀ ashamed | negative | NRC | disgust | 1058 words associated with disgust through MTurk crowdsourcing | only include negative words |
| hostile | hostile ǀ irritable | negative | General Inquirer | Arousal | 67 words indicating excitation; aside from pleasures or pains; but including arousal of affiliation and hostility | only include negative words |
| irritable | hostile ǀ irritable | negative | NRC | anger | 1247 words associated with anger through MTurk crowdsourcing | only include negative words |
| nervous | nervous ǀ jiittery | negative | LWIC | anxiety | 196 words associated with anxiety in the LWIC 2015 dictionary | only include negative words |
| jittery | nervous ǀ jiittery | negative | NRC | anticipation | 839 words associated with anticipation through MTurk crowdsourcing | only include negative words |
| afraid | afraid ǀ scared | negative | NRC | fear | 1476 words associated with fear through MTurk crowdsourcing | only include negative words |
| scared | afraid ǀ scared | negative | NRC | surprise | 534 words associated with surprise through MTurk crowdsourcing | only include negative words |

## Appendix B. Tests for Statistically Significant Differences per PANAS Group

The following four sub-sections within this section of the appendix provide the data for all of the statistically significant tests conducted between the ChatGPT-generated narratives and real tweets.

*Appendix B.1. Birth Narrative Comparison—Tweets versus ChatGPT-Generated Narratives*

Table A2 provides the results of the tests for significant differences between real tweets (sample 1) and ChatGPT-generated narratives (sample 2) for *Birth* life events. For rows displaying a value for $\chi^2$, the chi-squared test was applied to assess significance; otherwise, Fisher's exact test was applied. A null hypothesis is provided within the context of the tested components within the "Null Hypothesis Description" column and the resulting interpretation of the test based on a *p*-value assessment at an alpha level $\alpha = 0.05$ is provided in the "Interpretation" column.

*Appendix B.2. Death Narrative Comparison—Tweets versus ChatGPT-Generated Narratives*

Table A3 provides the results of the tests for significant differences between real tweets (sample 1) and ChatGPT-generated narratives (sample 2) for *Death* life events. For rows displaying a value for $\chi^2$, the chi-squared test was applied to assess significance; otherwise, Fisher's exact test was applied. A null hypothesis is provided within the context of the tested components within the "Null Hypothesis Description" column and the resulting interpretation of the test based on a *p*-value assessment at an alpha level $\alpha = 0.05$ is provided in the "Interpretation" column.

*Appendix B.3. Hired Narrative Comparison—Tweets versus ChatGPT-Generated Narratives*

Table A4 provides the results of the tests for significant differences between real tweets (sample 1) and ChatGPT-generated narratives (sample 2) for *Hired* life events. For rows displaying a value for $\chi^2$, the chi-squared test was applied to assess significance; otherwise, Fisher's exact test was applied. A null hypothesis is provided within the context of the tested components within the "Null Hypothesis Description" column and the resulting interpretation of the test based on a *p*-value assessment at an alpha level $\alpha = 0.05$ is provided in the "Interpretation" column.

*Appendix B.4. Fired Narrative Comparison—Tweets versus ChatGPT-generated Narratives*

Table A5 provides the results of the tests for significant differences between real tweets (sample 1) and ChatGPT-generated narratives (sample 2) for *Fired* life events. For rows displaying a value for $\chi^2$, the chi-squared test was applied to assess significance; otherwise, Fisher's exact test was applied. A null hypothesis is provided within the context of the tested components within the "Null Hypothesis Description" column and the resulting interpretation of the test based on a *p*-value assessment at an alpha level $\alpha = 0.05$ is provided in the "Interpretation" column.

**Table A2.** Tests for statistically significant differences in PANAS trait prevalence between real tweets and ChatGPT-generated narratives for *Birth* life events.

| PANAS_Group | *p* Value | Chi Square | Sample 1 Mean | Sample 1 Variance | Sample 2 Mean | Sample 2 Variance | Null Hypothesis Description | Interpretation |
|---|---|---|---|---|---|---|---|---|
| binary_positive | $1.54 \times 10^{-278}$ | 1271.77609751028 | 0.933046902971715 | 0.06248156521243 | 0.513771186440678 | 0.250075264662006 | There is no difference in association between ChatGPT and Twitter for event type Birth in relation to PANAS group: binary_positive. | Reject the null hypothesis. |
| binary_negative | $1.28 \times 10^{-1}$ | 2.3175529292593 | 0.410490511994271 | 0.242031379766721 | 0.4375 | 0.246354718981972 | There is no difference in association between ChatGPT and Twitter for event type Birth in relation to PANAS group: binary_negative. | Fail to reject the null hypothesis. |
| binary_interested_attentive_alert | $3.85 \times 10^{-85}$ | 382.341055827289 | 0.526494808449696 | 0.249342662193276 | 0.182203389830508 | 0.149163326563258 | There is no difference in association between ChatGPT and Twitter for event type Birth in relation to PANAS group: binary_interested_attentive_alert. | Reject the null hypothesis. |
| binary_excited_enthusiastic_inspired | $7.54 \times 10^{-276}$ | 1259.39464167701 | 0.841031149301826 | 0.133721693926593 | 0.310381355932203 | 0.214271752610673 | There is no difference in association between ChatGPT and Twitter for event type Birth in relation to PANAS group: binary_excited_enthusiastic_inspired. | Reject the null hypothesis. |
| binary_proud_determined | $6.84 \times 10^{-174}$ | 790.3284693054 | 0.729323308270677 | 0.197446166894407 | 0.260593220338983 | 0.192888725128961 | There is no difference in association between ChatGPT and Twitter for event type Birth in relation to PANAS group: binary_proud_determined. | Reject the null hypothesis. |
| binary_strong_active | $2.03 \times 10^{-72}$ | 323.91325121501 | 0.63139992839241 | 0.232775730091311 | 0.31885593220339 | 0.217417141470604 | There is no difference in association between ChatGPT and Twitter for event type Birth in relation to PANAS group: binary_strong_active. | Reject the null hypothesis. |
| binary_distressed_upset | $1.27 \times 10^{-34}$ | 150.62480849508 | 0.0631936985320444 | 0.0592108548644921 | 0.18114406779661 | 0.148488191311537 | There is no difference in association between ChatGPT and Twitter for event type Birth in relation to PANAS group: binary_distressed_upset. | Reject the null hypothesis. |
| binary_guilty_ashamed | $5.39 \times 10^{-39}$ | 170.629609145227 | 0.0907626208378088 | 0.082539543641044 | 0.236228813559322 | 0.180616091809407 | There is no difference in association between ChatGPT and Twitter for event type Birth in relation to PANAS group: binary_guilty_ashamed. | Reject the null hypothesis. |
| binary_hostile_irritable | $3.37 \times 10^{-2}$ | 4.50908186021549 | 0.361618331543144 | 0.230891847857269 | 0.32521186440678 | 0.219681821449755 | There is no difference in association between ChatGPT and Twitter for event type Birth in relation to PANAS group: binary_hostile_irritable. | Reject the null hypothesis. |
| binary_nervous_jittery | $1.99 \times 10^{-58}$ | 259.704615446121 | 0.333691371285356 | 0.2223812504788 | 0.0741525423728814 | 0.0687267465894998 | There is no difference in association between ChatGPT and Twitter for event type Birth in relation to PANAS group: binary_nervous_jittery. | Reject the null hypothesis. |
| binary_afraid_scared | $1.11 \times 10^{-51}$ | 228.762692248594 | 0.0415324024346581 | 0.0398145895497152 | 0.168432203389831 | 0.140211325197261 | There is no difference in association between ChatGPT and Twitter for event type Birth in relation to PANAS group: binary_afraid_scared. | Reject the null hypothesis. |

**Table A3.** Tests for statistically significant differences in PANAS trait prevalence between real tweets and ChatGPT-generated narratives for *Death* life events.

| PANAS_Group | *p*-Value | Chi Square | Sample 1 Mean | Sample 1 Variance | Sample 2 Mean | Sample 2 Variance | Null Hypothesis Description | Interpretation |
|---|---|---|---|---|---|---|---|---|
| binary_positive | $2.27 \times 10^{-98}$ | 443.113978240721 | 0.825024437927664 | 0.144387342969351 | 0.513771186440678 | 0.250075264662006 | There is no difference in association between ChatGPT and Twitter for event type Death in relation to PANAS group: binary_positive. | Reject the null hypothesis. |
| binary_negative | $4.56 \times 10^{-8}$ | 29.8974363771416 | 0.534897360703812 | 0.248830821492837 | 0.4375 | 0.246354718981972 | There is no difference in association between ChatGPT and Twitter for event type Death in relation to PANAS group: binary_negative. | Reject the null hypothesis. |
| binary_interested_attentive_alert | $1.85 \times 10^{-13}$ | 54.1561060728244 | 0.300097751710655 | 0.210080162519387 | 0.182203389830508 | 0.149163326563258 | There is no difference in association between ChatGPT and Twitter for event type Death in relation to PANAS group: binary_interested_attentive_alert. | Reject the null hypothesis. |
| binary_excited_enthusiastic_inspired | $3.62 \times 10^{-110}$ | 497.329564193082 | 0.69188660801564 | 0.213221215141308 | 0.310381355932203 | 0.214271752610673 | There is no difference in association between ChatGPT and Twitter for event type Death in relation to PANAS group: binary_excited_enthusiastic_inspired. | Reject the null hypothesis. |
| binary_proud_determined | $4.56 \times 10^{-73}$ | 326.897373112863 | 0.580840664711632 | 0.243512394435225 | 0.260593220338983 | 0.192888725128961 | There is no difference in association between ChatGPT and Twitter for event type Death in relation to PANAS group: binary_proud_determined. | Reject the null hypothesis. |
| binary_strong_active | $6.55 \times 10^{-63}$ | 280.273129126093 | 0.613294232649071 | 0.237210792369938 | 0.31885593220339 | 0.217417141470604 | There is no difference in association between ChatGPT and Twitter for event type Death in relation to PANAS group: binary_strong_active. | Reject the null hypothesis. |
| binary_distressed_upset | $5.98 \times 10^{-45}$ | 197.905833824767 | 0.424437927663734 | 0.244338142166999 | 0.18114406779661 | 0.148488191311537 | There is no difference in association between ChatGPT and Twitter for event type Death in relation to PANAS group: binary_distressed_upset. | Reject the null hypothesis. |
| binary_guilty_ashamed | $1.01 \times 10^{-6}$ | 23.9027316013673 | 0.316520039100684 | 0.216377406471645 | 0.236228813559322 | 0.180616091809407 | There is no difference in association between ChatGPT and Twitter for event type Death in relation to PANAS group: binary_guilty_ashamed. | Reject the null hypothesis. |
| binary_hostile_irritable | $1.25 \times 10^{-9}$ | 36.8931130725158 | 0.231867057673509 | 0.178139552131251 | 0.32521186440678 | 0.219681821449755 | There is no difference in association between ChatGPT and Twitter for event type Death in relation to PANAS group: binary_hostile_irritable. | Reject the null hypothesis. |
| binary_nervous_jittery | $5.19 \times 10^{-2}$ | 3.777433589273 | 0.0946236559139785 | 0.0856867717124823 | 0.0741525423728814 | 0.0687267465894998 | There is no difference in association between ChatGPT and Twitter for event type Death in relation to PANAS group: binary_nervous_jittery. | Fail to reject the null hypothesis. |
| binary_afraid_scared | $5.56 \times 10^{-8}$ | 29.5095216189555 | 0.251026392961877 | 0.188048907203158 | 0.168432203389831 | 0.140211325197261 | There is no difference in association between ChatGPT and Twitter for event type Death in relation to PANAS group: binary_afraid_scared. | Reject the null hypothesis. |

**Table A4.** Tests for statistically significant differences in PANAS trait prevalence between real tweets and ChatGPT-generated narratives for *Hired* life events.

| PANAS_Group | *p*-Value | Chi Square | Sample 1 Mean | Sample 1 Variance | Sample 2 Mean | Sample 2 Variance | Null Hypothesis Description | Interpretation |
|---|---|---|---|---|---|---|---|---|
| binary_positive | $1.57 \times 10^{-15}$ | 63.5460828619453 | 0.650082614283092 | 0.227516978116704 | 0.513771186440678 | 0.250075264662006 | There is no difference in association between ChatGPT and Twitter for event type Hired in relation to PANAS group: binary_positive. | Reject the null hypothesis. |
| binary_negative | $3.16 \times 10^{-15}$ | 62.1642168708749 | 0.306590783917753 | 0.212631911652103 | 0.4375 | 0.246354718981972 | There is no difference in association between ChatGPT and Twitter for event type Hired in relation to PANAS group: binary_negative. | Reject the null hypothesis. |
| binary_interested_attentive_alert | $1.73 \times 10^{-37}$ | 163.735966599973 | 0.40040389205067 | 0.240124699125503 | 0.182203389830508 | 0.149163326563258 | There is no difference in association between ChatGPT and Twitter for event type Hired in relation to PANAS group: binary_interested_attentive_alert. | Reject the null hypothesis. |
| binary_excited_enthusiastic_inspired | $5.59 \times 10^{-10}$ | 38.4587917449052 | 0.418211859739306 | 0.243355377068281 | 0.310381355932203 | 0.214271752610673 | There is no difference in association between ChatGPT and Twitter for event type Hired in relation to PANAS group: binary_excited_enthusiastic_inspired. | Reject the null hypothesis. |
| binary_proud_determined | $1.71 \times 10^{-14}$ | 58.8354788930908 | 0.39195887644575 | 0.238370877485921 | 0.260593220338983 | 0.192888725128961 | There is no difference in association between ChatGPT and Twitter for event type Hired in relation to PANAS group: binary_proud_determined. | Reject the null hypothesis. |
| binary_strong_active | $2.24 \times 10^{-16}$ | 67.379017495558 | 0.463190747200294 | 0.248690735367914 | 0.31885593220339 | 0.217417141470604 | There is no difference in association between ChatGPT and Twitter for event type Hired in relation to PANAS group: binary_strong_active. | Reject the null hypothesis. |
| binary_distressed_upset | $8.88 \times 10^{-10}$ | 37.5565339193202 | 0.11015237745548 | 0.0980368295128006 | 0.18114406779661 | 0.148488191311537 | There is no difference in association between ChatGPT and Twitter for event type Hired in relation to PANAS group: binary_distressed_upset. | Reject the null hypothesis. |
| binary_guilty_ashamed | $1.81 \times 10^{-13}$ | 54.1980526347543 | 0.141545805030292 | 0.121532902005443 | 0.236228813559322 | 0.180616091809407 | There is no difference in association between ChatGPT and Twitter for event type Hired in relation to PANAS group: binary_guilty_ashamed. | Reject the null hypothesis. |
| binary_hostile_irritable | $5.23 \times 10^{-12}$ | 47.5976419941658 | 0.221222691389756 | 0.172314847020812 | 0.32521186440678 | 0.219681821449755 | There is no difference in association between ChatGPT and Twitter for event type Hired in relation to PANAS group: binary_hostile_irritable. | Reject the null hypothesis. |
| binary_nervous_jittery | $3.56 \times 10^{-7}$ | 25.9203411667911 | 0.134202313199927 | 0.116213387633282 | 0.0741525423728814 | 0.0687267465894998 | There is no difference in association between ChatGPT and Twitter for event type Hired in relation to PANAS group: binary_nervous_jittery. | Reject the null hypothesis. |
| binary_afraid_scared | $3.66 \times 10^{-30}$ | 130.225462320604 | 0.0607673948962732 | 0.0570851987310565 | 0.168432203389831 | 0.140211325197261 | There is no difference in association between ChatGPT and Twitter for event type Hired in relation to PANAS group: binary_afraid_scared. | Reject the null hypothesis. |

**Table A5.** Tests for statistically significant differences in PANAS trait prevalence between real tweets and ChatGPT-generated narratives for *Fired* life events.

| PANAS_Group | *p*-Value | Chi Square | Sample 1 Mean | Sample 1 Variance | Sample 2 Mean | Sample 2 Variance | Null Hypothesis Description | Interpretation |
|---|---|---|---|---|---|---|---|---|
| binary_positive | $6.91 \times 10^{-1}$ | 0.157707980552807 | 0.506150174407931 | 0.250008073660913 | 0.513771186440678 | 0.250075264662006 | There is no difference in association between ChatGPT and Twitter for event type Fired in relation to PANAS group: binary_positive. | Fail to reject the null hypothesis. |
| binary_negative | $7.91 \times 10^{-36}$ | 156.134224715704 | 0.651918487240683 | 0.226962440655221 | 0.4375 | 0.246354718981972 | There is no difference in association between ChatGPT and Twitter for event type Fired in relation to PANAS group: binary_negative. | Reject the null hypothesis. |
| binary_interested_attentive_alert | $8.38 \times 10^{-11}$ | 42.1673533305469 | 0.284376721130898 | 0.203543969696702 | 0.182203389830508 | 0.149163326563258 | There is no difference in association between ChatGPT and Twitter for event type Fired in relation to PANAS group: binary_interested_attentive_alert. | Reject the null hypothesis. |
| binary_excited_enthusiastic_inspired | $3.91 \times 10^{-2}$ | 4.25626862354168 | 0.277033229300532 | 0.200322595847502 | 0.310381355932203 | 0.214271752610673 | There is no difference in association between ChatGPT and Twitter for event type Fired in relation to PANAS group: binary_excited_enthusiastic_inspired. | Reject the null hypothesis. |
| binary_proud_determined | $4.53 \times 10^{-2}$ | 4.00823113260934 | 0.293188911327336 | 0.207267225231407 | 0.260593220338983 | 0.192888725128961 | There is no difference in association between ChatGPT and Twitter for event type Fired in relation to PANAS group: binary_proud_determined. | Reject the null hypothesis. |
| binary_strong_active | $8.69 \times 10^{-1}$ | 0.0272449639570703 | 0.322195704057279 | 0.218425732533873 | 0.31885593220339 | 0.217417141470604 | There is no difference in association between ChatGPT and Twitter for event type Fired in relation to PANAS group: binary_strong_active. | Fail to reject the null hypothesis. |
| binary_distressed_upset | $3.20 \times 10^{-69}$ | 309.235428470664 | 0.489810905085368 | 0.249942068533279 | 0.18114406779661 | 0.148488191311537 | There is no difference in association between ChatGPT and Twitter for event type Fired in relation to PANAS group: binary_distressed_upset. | Reject the null hypothesis. |
| binary_guilty_ashamed | $1.17 \times 10^{-18}$ | 77.7566103482615 | 0.386451257572976 | 0.237150220860978 | 0.236228813559322 | 0.180616091809407 | There is no difference in association between ChatGPT and Twitter for event type Fired in relation to PANAS group: binary_guilty_ashamed. | Reject the null hypothesis. |
| binary_hostile_irritable | $9.81 \times 10^{-4}$ | 10.8626035628716 | 0.382045162474757 | 0.236130006773785 | 0.32521186440678 | 0.219681821449755 | There is no difference in association between ChatGPT and Twitter for event type Fired in relation to PANAS group: binary_hostile_irritable. | Reject the null hypothesis. |
| binary_nervous_jittery | $1.49 \times 10^{-29}$ | 127.442104037455 | 0.237929135303837 | 0.181352155829274 | 0.0741525423728814 | 0.0687267465894998 | There is no difference in association between ChatGPT and Twitter for event type Fired in relation to PANAS group: binary_nervous_jittery. | Reject the null hypothesis. |
| binary_afraid_scared | $2.47 \times 10^{-2}$ | 5.04651465849531 | 0.200477326968974 | 0.160315600247866 | 0.168432203389831 | 0.140211325197261 | There is no difference in association between ChatGPT and Twitter for event type Fired in relation to PANAS group: binary_afraid_scared. | Reject the null hypothesis. |

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
