# Peer review of "A Structured Narrative Prompt for Prompting Narratives from Large Language Models: Sentiment Assessment of ChatGPT-Generated Narratives and Real Tweets"

_futureinternet, doi:10.3390/fi15120375_

Round 1

Reviewer 1 Report

Comments and Suggestions for Authors

General comments

=============

The manuscript titled "A Structured Narrative Prompt for Large Language Models to Create Pertinent Narratives of Simulated Agents’ Life Events: A Sentiment Analysis Comparison" is a laudable attempt to demonstrate the applications of Large Language Models (LLMs) in generating narratives for simulated agents. The research trajectory is commendable; however, to elevate the clarity, rigor, and impact of this study, several areas of the paper could benefit from further clarification and expansion.

Specific comments

=============

Major comments

---------------------

[Title]

1. The title should encapsulate the main theme of the paper. Mentioning "ChatGPT" instead of the more generic "LLMs" would offer readers a clearer picture of the primary focus.

[Introduction]

2. Citations are crucial to establish the foundation of any scientific discourse. The specified lines lack proper referencing. Not only does this compromise the paper’s credibility, but it also deprives the reader of context.

3. Elaborating on why LLMs, particularly ChatGPT, were favored over other models is critical to justify the study's methodology. It also gives the reader insights into the unique capabilities of the selected models.

4. The medical applicability of LLMs needs elucidation. How does this limitation affect or contribute to the research in question? 

5. Clearly stating the research gaps and the hypothesis is pivotal. Readers should have a clear understanding of what was not known previously and what this study aims to prove or investigate.

[Methods]

6. The rationale behind model and tool selection is imperative for scientific rigor. Explicitly discussing the reasoning will help readers appreciate the methodology's nuances.

7. Justification for the chosen life events will allow readers to comprehend their significance in the study’s context.

8. Contextual background about LLMs, especially their healthcare applications, is more suited for the introduction. It helps set the stage for the study.

9. The 'archive of tweets' needs more elaboration. What does it entail, and why was it chosen?

10. Clarification on the two-by-two contingency analysis is required to ensure methodological transparency.

11. Providing a rationale for setting a significance level (P < 0.05) will guide readers about the study's statistical underpinnings.

12. Offering comparative reasons for choosing ChatGPT over tools like Google Bard would be insightful.

13. Stating the 'temperature' and 'n' settings, along with their justifications, will aid in the replicability of this study.

14. The iterative steps and field selections need more detailed elucidation for methodological robustness.

15. A clear description of how narratives were generated from Java classes is essential for comprehensibility.

16. A flow chart would provide a succinct visual representation of the study's methodology.

17. Elaborate on the evaluation techniques of the PANAS lexicon, highlighting the evaluators' role.

[Results]

18. Including statistical values in the results strengthens the narrative and allows for a deeper understanding of the findings.

19. The inclusion of P-values in figures 5-6 will offer a statistical foundation to the presented data.

[Discussion]

20. A distinct conclusion will encapsulate the study's findings and implications succinctly.

=============

Minor comments

---------------------

21. Stay updated with current terminologies. Hence, the renaming of Twitter should be acknowledged.

22. Distinctions between 'event' and 'life event' are required to avoid any ambiguity.

23. The terms "LLM" and "ChatGPT" need clear demarcation. Their interchangeability in the manuscript can be misleading.

24. A consistent referencing style will enhance the paper's readability and professionalism.

25. Ensure that abbreviations like "LaMDA" and "LLaMA" are used consistently to prevent confusion.

26. Semantics matter. Referring to ChatGPT as "aware" or "knowledgeable" anthropomorphizes the tool. Stick to neutral, accurate descriptors.

The paper holds promise, and with the above-mentioned modifications, it can stand as a solid contribution to the evolving landscape of AI-driven narrative generation.

Comments on the Quality of English Language

Please see the previous quality of English comments.

Author Response

Comments and Suggestions for Authors

General comments

=============

The manuscript titled "A Structured Narrative Prompt for Large Language Models to Create Pertinent Narratives of Simulated Agents’ Life Events: A Sentiment Analysis Comparison" is a laudable attempt to demonstrate the applications of Large Language Models (LLMs) in generating narratives for simulated agents. The research trajectory is commendable; however, to elevate the clarity, rigor, and impact of this study, several areas of the paper could benefit from further clarification and expansion.

Thank you for your extensive and insightful comments. They have helped us to improve the quality of our article. Our point-by-point responses to each comment are provided below. In addition to the comments below, we have removed the Java-class generated narratives from our analysis and we have expanded our data set of ChatGPT-generated narratives for each of the four event types to better support our analysis and results.

Specific comments

=============

Major comments

---------------------

[Title]

  1. The title should encapsulate the main theme of the paper. Mentioning "ChatGPT" instead of the more generic "LLMs" would offer readers a clearer picture of the primary focus.

Thank you for your suggestion. The title has been updated to “A Structured Narrative Prompt for Prompting Narratives from Large Language Models: Sentiment Assessment of ChatGPT-generated Narratives and Real Tweets”

[Introduction]

  1. Citations are crucial to establish the foundation of any scientific discourse. The specified lines lack proper referencing. Not only does this compromise the paper’s credibility, but it also deprives the reader of context.

Thank you for your recommendation. Over 20 additional references have been added to the article and the Introduction has been largely rewritten to include more citations.

  1. Elaborating on why LLMs, particularly ChatGPT, were favored over other models is critical to justify the study's methodology. It also gives the reader insights into the unique capabilities of the selected models.

Thank you for your recommendation. Section 2.2 (LLM Selection) has been extended to better expand upon this point and justify our selection and utilization of ChatGPT-3.5.

  1. The medical applicability of LLMs needs elucidation. How does this limitation affect or contribute to the research in question?

Thank you for your comment. The Introduction has been reworked and less focus has been focused on the medical connections to LLMs. Justification has also been included in the Conclusion.

  1. Clearly stating the research gaps and the hypothesis is pivotal. Readers should have a clear understanding of what was not known previously and what this study aims to prove or investigate.

Thank you for your recommendation. The Introduction has been largely rewritten to better situate the study, specify the hypotheses, and convey the aims of the study.

[Methods]

  1. Justification for the chosen life events will allow readers to comprehend their significance in the study’s context.

Thank you for your recommendation. The Introduction has been largely rewritten to address this point. Also, a discussion of Agent Based Models has been added to better situate the study.

  1. Contextual background about LLMs, especially their healthcare applications, is more suited for the introduction. It helps set the stage for the study.

Thank you for your recommendation. The Introduction has been largely rewritten and a short discussion on LLMs is now included at the onset of the article. Justification has been added to the Background and Conclusion as well.

  1. The 'archive of tweets' needs more elaboration. What does it entail, and why was it chosen?

Thank you for your suggestion. We have expanded the final paragraph of Section 2.5 to further expand upon the description of the tweets. A reference has also been added to the IRB documentation for how the tweets were collected.

  1. Clarification on the two-by-two contingency analysis is required to ensure methodological transparency.

Thank you for the suggestion. Clarification has been added to the “Materials and Methods” section as well as to the “Results” section.

  1. Providing a rationale for setting a significance level (P < 0.05) will guide readers about the study's statistical underpinnings.

Thank you for your comment. The “Materials and Methods” section has been updated to reflect the selected significance level. Additionally, a limitation has been added to the “Limitations” section that describes why the significance level is appropriate and why a Bonferroni correction is not necessary based on the study design.

  1. Offering comparative reasons for choosing ChatGPT over tools like Google Bard would be insightful.

Thank you for your comment. Section 2.2 has been updated with additional content, references, and justification for choosing GPT-3.5.

  1. Stating the 'temperature' and 'n' settings, along with their justifications, will aid in the replicability of this study.

Thank you for your recommendation. We have added the utilized values for temperature and n as well as noted the default settings for both parameters. The role of these settings in the context of narrative generation is described in the “Lessons Learned” section.

  1. The iterative steps and field selections need more detailed elucidation for methodological robustness.

Thank you for your suggestion. Section 3.1 “LLM Structured Prompt for Narrative Generation” has two subsections broken up now that capture the path taken through the development of the Structured Narrative Prompt (Section 3.1.1) and the final LLM Narrative Prompt Structure (Section 3.1.2).

  1. A clear description of how narratives were generated from Java classes is essential for comprehensibility.

Thank you for your comment. We generated additional data and conducted additional tests on the data sets. The sample sizes for the JavaClass generated narratives were still smaller than desired. For robustness of the study, we decided to cut the JavaClass generated narratives. Th article has been completely reframed to only focus on ChatGPT-narratives to tweets comparisons.

  1. A flow chart would provide a succinct visual representation of the study's methodology.

Thank you for your comment. We quite agree. Figure 1 has been modified to more clearly convey its main points. Figure 2 has been added to provide the low level details on how the analysis is conducted.

  1. Elaborate on the evaluation techniques of the PANAS lexicon, highlighting the evaluators' role.

Thank you for your comment. We have expanded the first paragraph of Section 2.6 to further explain the evaluation of the PANAS lexicon. Also, the newly added Figure 2 breaks down this process to an algorithmic level.

[Results]

  1. Including statistical values in the results strengthens the narrative and allows for a deeper understanding of the findings.

Thank you for this recommendation. Statistical values have been integrated into the “Results” section. Tables A2-A5 have been updated to showcase the updated values for the expanded data sets. Figures 6-10 have been updated to showcase variation, delta values between the 2 group means, and p-values.

  1. The inclusion of P-values in figures 5-6 will offer a statistical foundation to the presented data.

Thank you for your comment. Section 3.2 is updated to add p-values to figures 7-10 that include p-values, standard-deviation bars, and delta values (for ChatGPT narratives and tweets, for each PANAS category binary analysis).

[Discussion]

  1. A distinct conclusion will encapsulate the study's findings and implications succinctly.

Thank you for your comment. We have added a “Conclusion” section and described the study’s findings.

=============

Minor comments

---------------------

  1. Stay updated with current terminologies. Hence, the renaming of Twitter should be acknowledged.

Thank you for your comment. We have acknowledged the change to X. Also, we have modified usages od the term Twitter and tweets to maintain consistency of usage throughout the article.

  1. Distinctions between 'event' and 'life event' are required to avoid any ambiguity.

Thank you for your comment. We have adjusted the text for consistency.

  1. The terms "LLM" and "ChatGPT" need clear demarcation. Their interchangeability in the manuscript can be misleading.

Thank you for your comment. We have modified the usages of the terms within the text for consistency and clarify.

  1. A consistent referencing style will enhance the paper's readability and professionalism.

Thank you for pointing this out. The article has been modified to get consistency in the in-text citation formatting. DOI’s have also been added to the reference list where missing from articles.

  1. Ensure that abbreviations like "LaMDA" and "LLaMA" are used consistently to prevent confusion.

Thank you for this comment. We have checked and modified our abbreviations of LLM platform terms for consistency and correctness.

  1. Semantics matter. Referring to ChatGPT as "aware" or "knowledgeable" anthropomorphizes the tool. Stick to neutral, accurate descriptors.

Thank you for pointing this out. We have modified the write-up for subjective usage of terms pertaining to ChatGPT.

The paper holds promise, and with the above-mentioned modifications, it can stand as a solid contribution to the evolving landscape of AI-driven narrative generation.

Thank you very much for your inciteful comments. They have led to a large rewrite of the article and an overall improvement of its contents.

Reviewer 2 Report

Comments and Suggestions for Authors

The authors conducted a study in which they compared the output of a Language Model (LLM) to narratives generated by an Agent-Based Model (ABM) and real tweets. They used sentiment analysis to assess the polarity of the text.

There are two key issues with the study design: First, the narratives generated by the LLM and the simulated ABM narratives were based on four specific event types (birth, death, hired, fired), while the real tweets covered a wide range of unrelated situations. This mismatch in the content of the narratives makes it challenging to compare the polarity of the tweets.

Secondly, the statistical analysis in the study may be overstating the significance of the differences observed. This is because the study involved many comparisons, and when conducting multiple tests, it's essential to apply the Bonferroni correction to adjust the significance level appropriately.

Additionally, a reference is missing on line 156.

The concept of the Agent-Based Model (ABM) is not adequately defined or referenced in the literature.

Finally, some sections of the paper are overly verbose, such as section 2.3, which discusses the usage of the ChatGPT API.

Author Response

Comments and Suggestions for Authors

Thank you for your insightful comments. They have helped us to improve the quality of our article. Our point-by-point responses to each comment are provided below. In addition to the comments below, we have removed the Java-class generated narratives from our analysis and we have expanded our data set of ChatGPT-generated narratives for each of the four event types to better support our analysis and results.

The authors conducted a study in which they compared the output of a Language Model (LLM) to narratives generated by an Agent-Based Model (ABM) and real tweets. They used sentiment analysis to assess the polarity of the text.

There are two key issues with the study design:

  1. First, the narratives generated by the LLM and the simulated ABM narratives were based on four specific event types (birth, death, hired, fired), while the real tweets covered a wide range of unrelated situations. This mismatch in the content of the narratives makes it challenging to compare the polarity of the tweets.

Thank you for the comment. We have made updates to better address this question within the article. The final paragraph of Section 2.4 (Data Sets) has been added to expand upon this topic. Item 7 in Section 4.2 (Limitations) also mentions this issue with respect to the generalizability of the findings. The goal of using an unrefined tweet dataset is to achieve a large, unbiased, vocabulary for comparison against the ChatGPT messages.

  1. Secondly, the statistical analysis in the study may be overstating the significance of the differences observed. This is because the study involved many comparisons, and when conducting multiple tests, it's essential to apply the Bonferroni correction to adjust the significance level appropriately.

Thank you for your recommendation. We agree that the Bonferroni correction is a commonly applied approach for adjusting significance level when conducting multiple tests; however, this test is not always recommended based on the design of the study. The article “Armstrong, R. A. (2014). When to use the Bonferroni correction. Ophthalmic and Physiological Optics, 34(5), 502-508.” provides several recommendations for when the use of the Bonferroni correction should and should not be considered. Based on this article, no correction is advised if the study is (1) restricted to a small number of planned comparisons, (2) an exploratory study of post-hoc testing of unplanned comparisons for further investigation, (3) if multiple simple tests are envisaged and it is the results of the individual tests that are important – in this case, the p-value for each individual test should be quoted and discussed, and (4) if it is imperative to avoid a type II error. A Bonferroni correction should be considered if (A) a single test of the universal null hypothesis that all tests are not significant is required, (B) it is imperative to avoid a type I error, and (C) a large number of tests are carried out without preplanned hypotheses in an attempt to establish that any result may be significant.

With respect to when the Bonferroni correction is not advised, items 1, 2, 3, and 4 apply to our study. With respect to when the Bonferroni is advised, none of the items A, B, and C apply to the design of our study. Therefore, we do not believe that applying a Bonferroni correction to our results is an appropriate path for us to take with our analysis. Additionally, we have reduced and rerun our analysis to only evaluate statistically significant differences between the tweets and the ChatGPT-generated narratives. This reduces the number of comparisons from 132 to 44.

Since we are interested in the individual comparisons for each PANAS category, not applying the Bonferroni correction is more appropriate for the design of our study. This is often the case in exploratory research where the goal is to identify interesting patterns or potential relationships that may warrant further investigation. In this scenario, we are willing to accept a higher risk of Type I errors (false positives) to reduce the chance of Type II errors (false negatives). This means that we are more willing to identify something as significant even if it might not be, in order to not miss any potential findings.

We have added this explanation to Item 8 of our Limitations (Not Correcting for Multiple Comparisons) so that this is also stated explicitly for the readers.

  1. Additionally, a reference is missing on line 156.

Thank you for pointing this out. In Section 2, the noted in-text citation has been added for the tweet source: Gore and Lynch 2022 Understanding Twitter Users.

  1. The concept of the Agent-Based Model (ABM) is not adequately defined or referenced in the literature.

Thank you for pointing this out. This assumption was an oversight and background has been added on Agent Based Modeling along with numerous supporting references.

  1. Finally, some sections of the paper are overly verbose, such as section 2.3, which discusses the usage of the ChatGPT API.

Thank you for your suggestion. We have rewritten most sections of the article to condense text for the purpose of conciseness and clarity.

Round 2

Reviewer 1 Report

Comments and Suggestions for Authors

As a reviewer, I would like to commend the authors for their manuscript titled "A Structured Narrative Prompt for Prompting Narratives from Large Language Models: Sentiment Assessment of ChatGPT-generated Narratives and Real Tweets."

The manuscript offers an insightful exploration into the capabilities of Large Language Models (LLMs) in constructing meaningful narratives that simulate life events of agents, which is a significant step forward in narrative generation research. Almost all responses were reasonable except of the following format of the p-value.

Regarding the format of the p-value reported in the manuscript and particularly in figures 8 and 9, it is customary in scientific communication to present p-values less than 0.001 simply as "p < 0.001." This notation is preferred over the current format of "3.55x10^-07" or "0.000," as it provides a clear, standardized, and easily interpretable indication that the results are highly statistically significant without delving into the exact decimals, which can be cumbersome and unnecessary for the reader. Adopting this convention would align your reporting with commonly accepted practices and enhance the manuscript's professionalism and readability.

Author Response

Comments and Suggestions for Authors

As a reviewer, I would like to commend the authors for their manuscript titled "A Structured Narrative Prompt for Prompting Narratives from Large Language Models: Sentiment Assessment of ChatGPT-generated Narratives and Real Tweets."

Thank you for the time and effort that you put into reviewing our article. We appreciate all of your suggestions throughout both review rounds and are article is improved as a result.

The manuscript offers an insightful exploration into the capabilities of Large Language Models (LLMs) in constructing meaningful narratives that simulate life events of agents, which is a significant step forward in narrative generation research. Almost all responses were reasonable except of the following format of the p-value.

Regarding the format of the p-value reported in the manuscript and particularly in figures 8 and 9, it is customary in scientific communication to present p-values less than 0.001 simply as "p < 0.001." This notation is preferred over the current format of "3.55x10^-07" or "0.000," as it provides a clear, standardized, and easily interpretable indication that the results are highly statistically significant without delving into the exact decimals, which can be cumbersome and unnecessary for the reader. Adopting this convention would align your reporting with commonly accepted practices and enhance the manuscript's professionalism and readability.

Thank you for pointing this out. Figures 7-10 have been updated accordingly.

Reviewer 2 Report

Comments and Suggestions for Authors

Most of the issues exposed by the first review has been solved. The paper is clearer now and the justification about not applying the Bonferroni correction is appropriate.

However, the comparison between real tweets and LLM-generated narratives is not justified: as the real tweets are not related to the 4 events of the ABM, it is not clear how and why they are compared to each one of these categories. Are all the real tweets compared to each one of the events or are they randomly split into four groups?

However, the article is still too long and cumbersome, making it difficult to follow the arguments. Some points should be reduced or eliminated to focus on the main hypothesis of the work.

The dataset repository (ref. 86) is not available.

Author Response

Comments and Suggestions for Authors

Most of the issues exposed by the first review has been solved. The paper is clearer now and the justification about not applying the Bonferroni correction is appropriate.

Thank you for the time and effort that you put into reviewing our article. We appreciate all of your suggestions throughout both review rounds and are article is improved as a result.

However, the comparison between real tweets and LLM-generated narratives is not justified: as the real tweets are not related to the 4 events of the ABM, it is not clear how and why they are compared to each one of these categories. Are all the real tweets compared to each one of the events or are they randomly split into four groups?

The ChatGPT-generated narratives are compared to the existing archive of tweets in order to attain a static sentiment value for each PANAS trait that can be compared against each corresponding sentiment value from the generated narratives. Section 2.1 “Sentiment Scoring and Sentiment Analysis” has been updated to better justify this comparison. For consistency and to reduce redundancy, related language from Section 2, Section 2.2, and Section 2.4 have been removed and incorporated within the updates for Section 2.1. The entire tweet set is compared to each of the four events. This also helps to showcase the differences in sentiment levels for the four event types driving the ChatGPT-generated narratives while maintaining a static baseline for comparison from the tweets.

However, the article is still too long and cumbersome, making it difficult to follow the arguments. Some points should be reduced or eliminated to focus on the main hypothesis of the work.

Thank you for your recommendation. We have made additional reductions to content throughout the article to further improve the article’s flow.

The dataset repository (ref. 86) is not available.

Thank you for pointing this out. The online repository has been published and is publicly accessible.